# An optoacoustic field-programmable perceptron for recurrent neural networks

Steven Becker[1,2], Dirk Englund [ID][3] & Birgit Stiller [ID][1,2] ✉

Recurrent neural networks (RNNs) can process contextual information such as time series signals and language. But their tracking of internal states is a limiting factor, motivating research on analog implementations in photonics. While photonic unidirectional feedforward neural networks (NNs) have demonstrated big leaps, bi-directional optical RNNs present a challenge: the need for a short-term memory that (i) programmable and coherently computes optical inputs, (ii) minimizes added noise, and (iii) allows scalability. Here, we experimentally demonstrate an optoacoustic recurrent operator (OREO) which meets (i, ii, iii). OREO contextualizes the information of an optical pulse sequence via acoustic waves. The acoustic waves link different optical pulses, capturing their information and using it to manipulate subsequent operations. OREO's all-optical control on a pulse-by-pulse basis offers simple reconfigurability and is used to implement a recurrent drop-out and pattern recognition of 27 optical pulse patterns. Finally, we introduce OREO as bi-directional perceptron for new classes of optical NNs.

Understanding the context of a situation is a powerful ability of the human brain, allowing it to predict possible outcomes and to make intelligent decisions. While humans can access the context of a situation via the short-term memory, machines struggle in contextualizing. Artificial neural networks, one of the most powerful computing architectures, face this problem as well. To overcome this limitation, they can be equipped with recurrent feedback, allowing them to process current inputs based on previous ones. The so-called recurrent neural networks (RNNs) can contextualize, recognize, and predict sequences of information and are applied for numerous applications such as language processing tasks, and for video and image processing[1–5]. One of the simplest versions of a RNN is the Elman network[6], which adds a recurrent operation to each neuron of its fully-connected network, analogous to the neuron's activation function. With this three-layer network, Elman was already able to understand simple grammatical structure. More complex models have proven themselves as Chinese poets, rap artists, and empathetic listeners[7–9].

Currently, the scientific community aims to transfer electronic neural networks into the optical domain. The resulting optical neural networks have attracted great interest due to their promises of high processing speed and broad bandwidth, and low dissipative losses[10–12]. Thus, they are considered to pave the way towards energy efficient and highly parallel optical circuits, enhancing the performance and capabilities of future artificial neural networks[13–19].

Although the field of optical neural networks has made great progress in recent years, the field of recurrent optical neural networks is still very limited to concepts based on artificial reservoirs, such as free-space cavities[20], delay systems[21,22], and microring resonators[23]. These designs can face several challenging issues. Firstly, the usage of an artificial cavity, e.g. a ring resonator, requires additional tuning of the individual rings due to manufacturing depended properties such as the free spectral range (FSR). This requires additional compensation routines in order to match the FSR witch the chosen temporal dynamic, wavelength and coupling. Secondly, the free-space cavities and delay systems may not be frequency sensitive, preventing them from being applied for resource-efficient multi-frequency data processing. Finally, the cavity's recurrent weights cannot be varied rapidly, limiting the control of the recurrent process such as the implementation of recurrent dropout on single pulse level in order to regularize the network.

[1]Max Planck Institute for the Science of Light, Staudtstr. 2, 91058 Erlangen, Germany. [2]Department of Physics, Friedrich-Alexander-Universität Erlangen-Nürnberg, Staudtstr. 7, 91058 Erlangen, Germany. [3]Research Laboratory of Electronics, Massachusetts Institute of Technology, Cambridge, MA 02139, USA. ✉e-mail: birgit.stiller@mpl.mpg.de

Here, we experimentally demonstrate an optoacoustic recurrent operator (OREO) based on stimulated Brillouin-Mandelstam scattering (SBS) that can unlock recurrent functionalities in existing optical neural network architectures (see Fig. 1a). SBS is an interaction of optical waves with traveling acoustic waves which serve in our system as a latency component due to the slow acoustic velocity. OREO is therefore able to contextualize a time-encoded stream of information by using acoustic waves as a memory to remember previous operations (see Fig. 1b).

In contrast to previously reported approaches[20-24], OREO controls its coherent recurrent operation completely optically on pulse level without the need of any artificial reservoir such as a ring resonator or a delay system. Hence, OREO does not rely on complicated manufacturing processes of microstructures. It functions in any optical waveguide, including on-chip devices, as it harvests the physical property of a sound wave[25-27]. With the announcement of the first on-chip EDFA[28] a fully integrated design is in reach. While recognizing the outstanding efforts of the scientific community to implement non-reciprocal devices on-chip[29-34], we would like to highlight the work[35] which implements an on-chip circulator free SBS experiment. More details about how OREO could be integrated on-chip can be found in the discussion.

We demonstrate OREO experimentally from different perspectives. Firstly, we show how OREO links different input states of subsequent optical pulses to each other via acoustic waves. Secondly, we present how the all-optical control of OREO can be used to implement a recurrent dropout. Finally, we apply OREO as an acceptor[36] to predict up to 27 different patterns carried by a time series of input pulses.

## Results

### Concept of an optoacoustic recurrent operator

The recurrent operation of OREO is based on the interaction of optical and acoustic waves through SBS, which is one of the most prominent third-order nonlinear effects and describes the coherent coupling of two optical waves, data and control, to an acoustic wave in a material. The

dynamic is illustrated in Fig. 1c and follows from the Hamiltonian (1)[37-39]:

$$H_{\text{OREO}} = \hbar\omega_D \int_{-\infty}^{\infty} dz\, a_D^\dagger(z,t) a_D(z,t) + \hbar\omega_C \int_{-\infty}^{\infty} dz\, a_C^\dagger(z,t) a_C(z,t) +$$
$$+ \hbar\Omega \int_{-\infty}^{\infty} dz\, b^\dagger(z,t) b(z,t) + \underbrace{\hbar g_0 \int_{-\infty}^{\infty} dz\, \left( a_D(z,t)\, a_C^\dagger(z,t)\, b^\dagger(z,t) + \text{H.c.} \right)}_{\text{Interaction Hamiltonian}},$$

(1)

using the optoacoustic coupling constant $g_0$, the frequency relation between the optical fields $\omega_D = \omega_C + \Omega$, and the wave packet operators $a_D$, $a_C$, $b$ of the data, control and acoustic field, respectively. Similar to the clinking of a wine glass, the acoustic wave $b$ persists beyond its excitation, decaying exponentially with time $b(t) \propto \exp(-t/\tau_{\text{ac}})$, where $\tau_{\text{ac}} \propto \Gamma_{\text{ac}}^{-1}$ is the acoustic lifetime, which depends on the properties of the used waveguide and is for a photonic crystal fiber (PCF) about $\tau_{\text{ac}} \approx 10$ ns (see Fig. 1c). As a result, an acoustic wave $b_i$ can seed subsequent SBS processes $j > i$. Moreover, the acoustic builds up with each SBS process, which can be described as a superposition of all previous created acoustic waves $b_i$ with amplitude $b_{0,i}$, created at the time $t_i$ and carrying a phase $\varphi_i$. Hence, the acoustic wave $b_N$ after $N$ SBS interactions:

$$b_N(z,t) = \sum_{i=1}^{N} b_i(z,t) = \sum_{i=1}^{N} b_{0,i}(z)\, e^{-\frac{t-t_i}{\tau_{\text{ac}}} + i\varphi_i(t)}, \quad (2)$$

yields the recurrence in the interaction Hamiltonian:

$$H_{\text{int,N}} = \hbar g_0 \int_{-\infty}^{\infty} dz\, \left( a_D(z,t)\, a_C^\dagger(z,t) \left( \sum_{i=1}^{N} b_i^\dagger(z,t) \right) + \text{H.c.} \right) \quad (3)$$

Equation (3) shows furthermore that programming the field $a_C$ controls the acoustic feedback all-optically, enabling a pulse-by-pulse increase or suppression. For instance, setting $a_{C,i}=0$ corresponds to a recurrent dropout so that $a_{D,i}$ leaves the fiber unchanged.

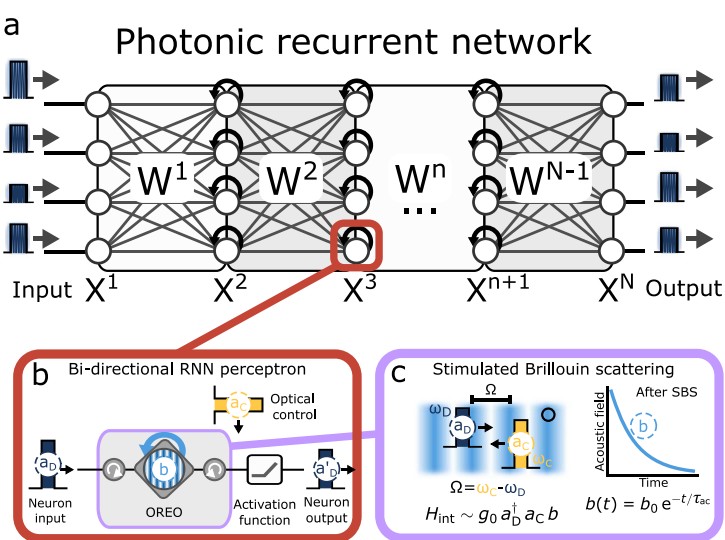

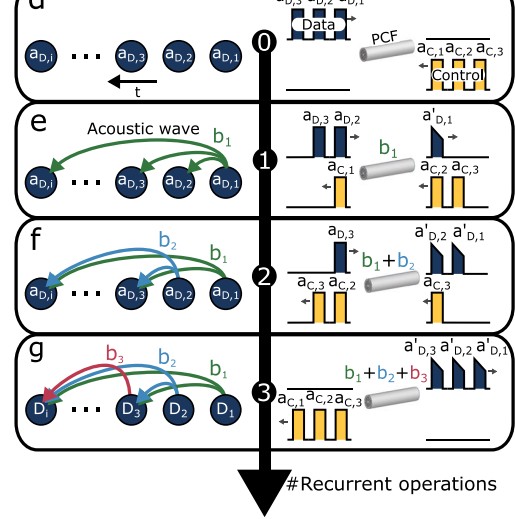

**Fig. 1 | Schematic of the optoacoustic recurrent operator (OREO) and its proposed function in a recurrent optical NN. a** An example of a photonic recurrent network with $N$ layers $X^n$, which are connected by a matrix operation $W^n$. **b** The bi-directional perceptron contains an OREO and an activation function. OREO captures and links sequential information $a_D$ using a sound wave $b$, which is generated by SBS and controlled by an optical control pulse $a_C$. The output of the acoustic recurrent neuron $a'_D$ is fed into the next layer of the optical neural network. The blue arrow indicates the recurrent nature of one neuron. **c** Conceptional illustration of the SBS process with its interaction Hamiltonian $H_{\text{int}}$. The sound wave $b$ carries the information of the neuron's input and decays

after the SBS process exponentially with the acoustic lifetime $\tau_{\text{ac}}$. **d–g** Illustration of three recurrent operations performed by OREO. **d** shows the initial situation with three data-control pulse pairs separated by a deadtime $dt$. The data and control pulses are launched from opposite sides into a photonic crystal fiber (PCF). **e** shows the system after the SBS-interaction of $a_{D,1}$ and $a_{C,1}$, which transfers energy from $a_{D,1}$ to an acoustic wave $b_1$. **f** shows the system after a second pulse pair has passed the PCF. The acoustic wave $b_1$ connects the interaction of $a_{D,2}$ and $a_{C,2}$ with the previous one, while the SBS process transfers information from $a_{D,2}$ into $b_2$. **g** highlights the acoustic link created by OREO between three optical pulse pairs.

We experimentally implement OREO in a telecom-fiber apparatus illustrated in Fig. 1d. Here we launch several consecutive rectangular optical input data pulses $a_{D,i}$ and strong counter-propagating optical control pulses $a_{C,i}$ into a PCF. The optical data pulses are shifted up in frequency by $\Omega/2\pi \approx 10.6$ GHz compared to the optical control pulses, which is close to the Brillouin frequency of the PCF. When a data and control pulse pair $a_{D,1}$ and $a_{C,1}$ meets inside the PCF, they induce SBS, depleting the data pulse and transferring its energy into the acoustic domain. Eventually, an acoustic wave $b_1$ is generated, which persists much longer than the optical interaction (see Fig. 1e). An optoacoustic recurrent operation is performed, when a subsequent data and control pulse pair ($a_{D,2}$ and $a_{C,2}$) reaches the acoustic wave $b_1$ before it has decayed. Hence, the deadtime $dt$ until the second pulse pair arrives must be less then the acoustic lifetime. The previously generated acoustic wave connects to the subsequent SBS process between $a_{D,2}$ and $a_{C,2}$ and establishes a link of the second data pulse $a_{D,2}$ to the first data pulse $a_{D,1}$. In addition, the second SBS process creates a second acoustic wave $b_2$, carrying information of $a_{D,2}$. Now, the acoustic domain holds information of both data pulses $a_{D,1}$ and $a_{D,2}$ (see Fig. 1f). The discussed procedure could now be repeated also for a third pulse pair and, in general, as long as subsequent pules pairs arrive before the acoustic wave decayed completely (see Fig. 1g).

## Recurrent operation

In the following, we continue with the previously introduced setting of three pulse pairs ($a_{D,i}, a_{C,i}$), $i=1,2,3$. We use this configuration to experimentally study the recurrent feedback, established by the acoustic waves $b_i$. Precisely, three different acoustic links occur. On the

one hand, $b_1$ connects the input $a_{D,1}$ to the output $a'_{D,2}$ and $a'_{D,3}$ and, on the other hand, $b_2$ connects $a_{D,2}$ to $a'_{D,3}$ (see Fig. 2a). In order to study the different connections, we sweep the input amplitude of either $a_{D,1}$ or $a_{D,2}$, while keeping the other data pulses constant. For instance, if the input amplitude of $a_{D,1}$ is varied, $a_{D,2}$ and $a_{D,3}$ are fixed in amplitude. The control pulses $a_{C,i}$ are kept constant over the entire study. For each amplitude step, we measure the area under the curve (AuC) of the output pulses $a'_{D,i}$. In the subsequent analysis, we normalize the AuC of $a'_{D,i}$ with the AuC of an input data pulse $a_{D,i}$ with an amplitude scale of 1, representing the highest input value. In order to measure the reference, we launch a data pulse sequence into OREO without counter-propagating control pulses. In order to rule out drifting effects, we measure each amplitude twice in a random order and take the mean value afterwards. Furthermore, the amplitude sweep is performed for three different time delays $dt = 2.5, 4.5,$ and $10$ ns, as the acoustic link decays over time. In order to have a reference for the dynamic of the Brillouin process, we use the interaction of a single data-control pulse pair. The data pulse $a_D$ gets depleted by the SBS process in this single pulse interaction (SPI). The degree of depletion depends mainly on the power carried by the control pulse[25]. For OREO, we deplete about 47% of the data pulse. For a deadtime of 2.5 ns, an increase in amplitude of $a_{D,1}$ raises the output amplitude $a'_{D,2}$ as shown by Fig. 2b. The reason for this dynamic is that $a_{D,2} \leftrightarrow a_{C,2}$ is seeded by the acoustic wave $b_1$ influencing the degree of depletion. More precisely, the dynamic can be explained with the different acoustic phases of $b_1$ and $b_2$ (see Eq. (2)), which can lead to constructive or destructive interference during the SBS process. The acoustic phase is introduced by detuning the frequency difference between data and control pulses

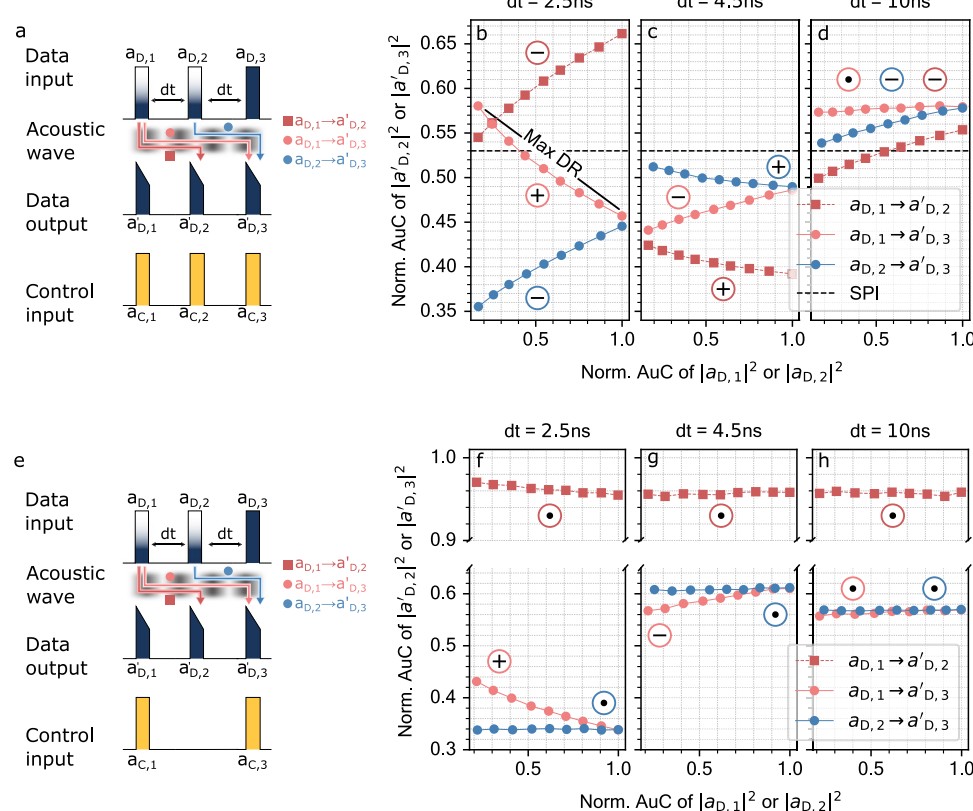

**Fig. 2 | Observing OREO's optoacoustic linking and recurrent dropout capabilities. a** Schematic illustration of the amplitude sweep that investigates how different optical states are passed between the optical data pulses $a_{D,i}$ via an acoustic wave $b$. **b–d** Experimental results of the amplitude sweep. While $a_{D,i}$, $i=1,2$ is changed, its impact on the subsequent pulses $a_{D,j}$, $j>i$ is studied for different deadtimes $dt$. Each SBS process creates an acoustic wave $b_i$, which

interferes with pre-existing ones $b_k$, $k<i$, eventually. We mark the links $a_{D,i} \to a_{D,j}$ with a +, −, and ·, when they experienced an enhancement, a reduction and an annihilation of the SBS process, respectively. We added the depletion of a single pulse interaction (SPI) as reference. **e** Schematic illustration of the pulse configuration used to study the OREO's feature to implement a recurrent dropout. **f–h** Experimental results of OREO's recurrent dropout capabilities.

slightly from the Brillouin frequency. For example, if the interaction $a_{D,i} \leftrightarrow a_{C,i}$ has created the acoustic wave $b_i$ with a specific phase (see Eq. (2)), then a second SBS process $j > i$ would generate dynamically $b_j$ on top of $b_i$. Both $b_i$ and $b_j$ have a different phase which lead to acoustic interference[40]. As Brillouin scattering describes a dynamic coupling between photons and phonons, the acoustic interference effects the overall "stimulated" dynamic. Hence, if $b_i$ and $b_j$ interfere destructively, the "stimulated" dynamic is slowed down and, therefore, the depletion of the data pulse $a_{Dj}$ lower. However, if $b_i$ and $b_j$ add up via constructive interference, then the stimulation is accelerated, increasing the depletion of $a_{Dj}$. The acoustic interference is also the reason for the decreasing behavior of the link $a_{D,1} \rightarrow a'_{D,3}$ on $a_{D,3} \leftrightarrow a_{C,3}$. Here, the acoustic wave $b_1$ accelerates the SBS process of $a_{D,3} \leftrightarrow a_{C,3}$ which leads to an increased depletion of $a_{D,3}$.

The symbols + and − mark the constructive and destructive nature of the underlying acoustic interference in Fig. 2b, respectively. OREO achieves a maximal dynamic range (Max DR) of 33%. For a deadtime of $dt = 4.5$ ns we observe a flip in the dynamic as all links switch their behavior from a constructive (+) to destructive (−) acoustic link, and vice versa (depicted in Fig. 2c). In addition, the overall level of depletion is larger in comparison to the SPI-case. For a deadtime of $dt = 10$ ns (equal to the acoustic lifetime), the dynamic range of the optical connection decreases further as we can see for the connection $a_{D,1} \rightarrow a'_{D,2}$ in Fig. 2d. Ultimately, the effect of the decaying acoustic wave becomes in particular visible for the interaction $a_{D,1} \rightarrow a'_{D,3}$ as $a'_{D,3}$ remains constant over the entire sweep range of $a_{D,1}$ (see Fig 2d). Note that we marked vanished acoustic links with the •-symbol.

With this, we have shown that OREO connects the information carried by subsequent optical data pulses. The acoustic link is sensitive to the amplitude and deadtime of the involved optical data pulses. As the interaction is continuous, it can be used for digital and analog recurrent tasks. Moreover, the acoustic interference observed with OREO ties in with previous studies based on continuous optical waves and our measurement extends the observation of acoustic interference into a pulsed context[40,41]. Being able to distinguish nine different amplitude levels, we conclude that OREO can resolve at least 3-bit information. In general, SBS is sensitive to continous amplitude levels, hence higher number of unique symbols can be encoded. The fundamental limitation is the data encoding and detection. In the supplementary material, we study OREO numerically and experimentally in a highly nonlinear fiber (HNLF), using the framework presented in ref. [42]. With the HNLF we study the linear response of OREO, which occurs in the case that the frequency difference of data and control matches exactly the Brillouin frequency. OREO controls the recurrent operation completely optically via the control pulses, enabling us to implement use case specific computations. For instance, in a pulse sequence consisting of three data pulses, one could skip the middle pulse $a_{D,2}$ by dropping the second control pulse, which could be useful for regularization[43]. In order to demonstrate the recurrent dropout, we excluded the second control pulse $a_{C,2}$ from the pulse train. Note, that the amplitudes of the other control pulses remain the same. In a next step, we vary the amplitude of data pulses $a_{D,1}$ and $a_{D,2}$ in upward and downward direction and check the impact on the subsequent data pulses (see Fig. 2e). Furthermore, we change the deadtime to investigate the influence of the acoustic interference on the interaction $a_{D,1} \rightarrow a'_{D,3}$.

OREO turns off the links between $a_{D,1} \rightarrow a'_{D,2}$ and $a_{D,2} \rightarrow a'_{D,3}$ as we can see in in Fig. 2f. As marked with the •-symbol, those two links show a constant behavior for the entire amplitude sweep. Only the interaction $a_{D,1} \rightarrow a'_{D,3}$ is active as the control pulses $a_{C,1}$ and $a_{C,3}$ establish the required acoustic link. Note, that for the case of $a_{D,2} \rightarrow a'_{D,3}$ the interaction $a_{D,3} \leftrightarrow a_{C,3}$ is influenced by the acoustic wave generated of the $a_{D,1} \leftrightarrow a_{C,1}$-interaction ($a_{D,1}$ is constant). This link can also explain the lower degree of depletion of $a_{D,1} \rightarrow a'_{D,3}$ at $dt = 4.5$ ns (see Fig. 2g). Here, the $a_{D,1}$ and $a_{D,3}$ are already separated by 10 ns,

which eliminated almost their acoustic link. At a deadtime of $dt = 10$ ns, the $a'_{D,3}$ is completely disconnected from $a_{D,1}$ and $a_{D,2}$ as can be seen by the constant behavior of $a'_{D,3}$ for both sweeps of $a_{D,1}$ and $a_{D,2}$ (see Fig. 2h). Besides, over all measurements, $a'_{D,2}$ is below the reference level ($a'_{D,2} < 1$), e.g., for the interaction $a_{D,1} \rightarrow a'_{D,2}$ in Fig. 2f. The increased optical noise floor appears as soon as the EDFA is turned on and could lead to this intrinsic depletion.

## Optical pattern recognition

From the beginning on, recurrent operators have been used to recognize patterns[6]. In the following section, we experimentally employ OREO as an acceptor[36] to recognize any pattern that can be created with two different data pulses $a$ and $b$: $aa$, $ab$, $ba$ & $bb$, where the $b$-pulse is half the amplitude of the $a$-pulse. Each pulse is launched with a matching control pulse $a_{C,i}$ into the PCF, where SBS is introduced. The deadtime between two consecutive pulses is 2.5 ns. Figure 3a shows schematically the information flow of the acceptor task. The individual pulses of the pattern $ab$ are launched into OREO each after 2.5 ns with a corresponding control pulse $a_{C,i}$. The information of the pattern is captured by the acoustic wave and flows in time, connecting the different SBS processes. Eventually, the sequential information captured by OREO is evaluated with a third optical evaluation pulse (Eval). In this way, we map the sequential information onto a single optical pulse which can be used then for classification, realizing experimentally a photonic acceptor. We perform the final classification by measuring the output evaluation pulse (Eval'), extracting its AuC in post-processing and then feeding this value into a digital Random Forest classifier[44] (RFC). In total, we check all patterns 250-times in a random order and classify the resulting experimentally obtained data set (70% training, 30% testing) with the RFC implemented in scikit-learn-package v1.1.3[45]. Furthermore, we perform the described study twice, once with the SBS-process and once without to isolate OREO's effect.

When OREO is off, the RFC cannot distinguish the different patterns and shows the same accuracy as a random guess (see Fig. 3b). However with OREO, the RFC is capable of distinguishing the different patterns almost with an accuracy of almost a 100% (see Fig. 3c).

Next, pushing OREO to the acoustic lifetime limit, we evaluate its performance for three different states encoded onto three pulses. The third state $c$ is three quarters of the $a$ state. In total, we test OREO to distinguish every possible permutation of $a$, $b$, and $c$, giving 27 different patterns. This time we launch a fourth data-control pair into the PCF, in order to evaluate OREO's memory (see Fig. 3d). Note that all four control pulses $a_{C,i}$ carry the same optical energy as in the three pulse configuration. We increased the sample size $n$ per pattern from 250 to 500 measurements in order to decrease statistical errors. Figure 3e shows the corresponding confusion matrix. OREO functions as acceptor and generates distinguishable distributions for the 27 patterns. The RFC achieves an accuracy of 45%, exceeding the accuracy of a simple guess by 11-times. The performance of OREO is currently limited by experimental precision, which is reduced by drifts of the optical pulses over the measurement period. Therefore, we perform a numerical analysis of OREO as an acceptor in the frequency matched case, in order to assess its potential performance. In this simulated experiment, OREO and the RFC achieve an accuracy of 92%. Figure 3f shows the corresponding confusion matrix. In the supplementary material, we describe the numerical analysis and check the impact of the pulse width, deadtime, acoustic lifetime, and experimental precision on OREO's pattern recognition performance. This analysis indicates that OREOs performance can even be pushed further to an accuracy of 97%. The current implementation of OREO can be seen as a photonic recurrent extreme learning machine with one layer and one neuron. Hence, we compare the result of the $abc$-study to a digital RNN-classifier containing a single recurrent neuron and a fully-connected network (FCN). This classifier achieves in the same task

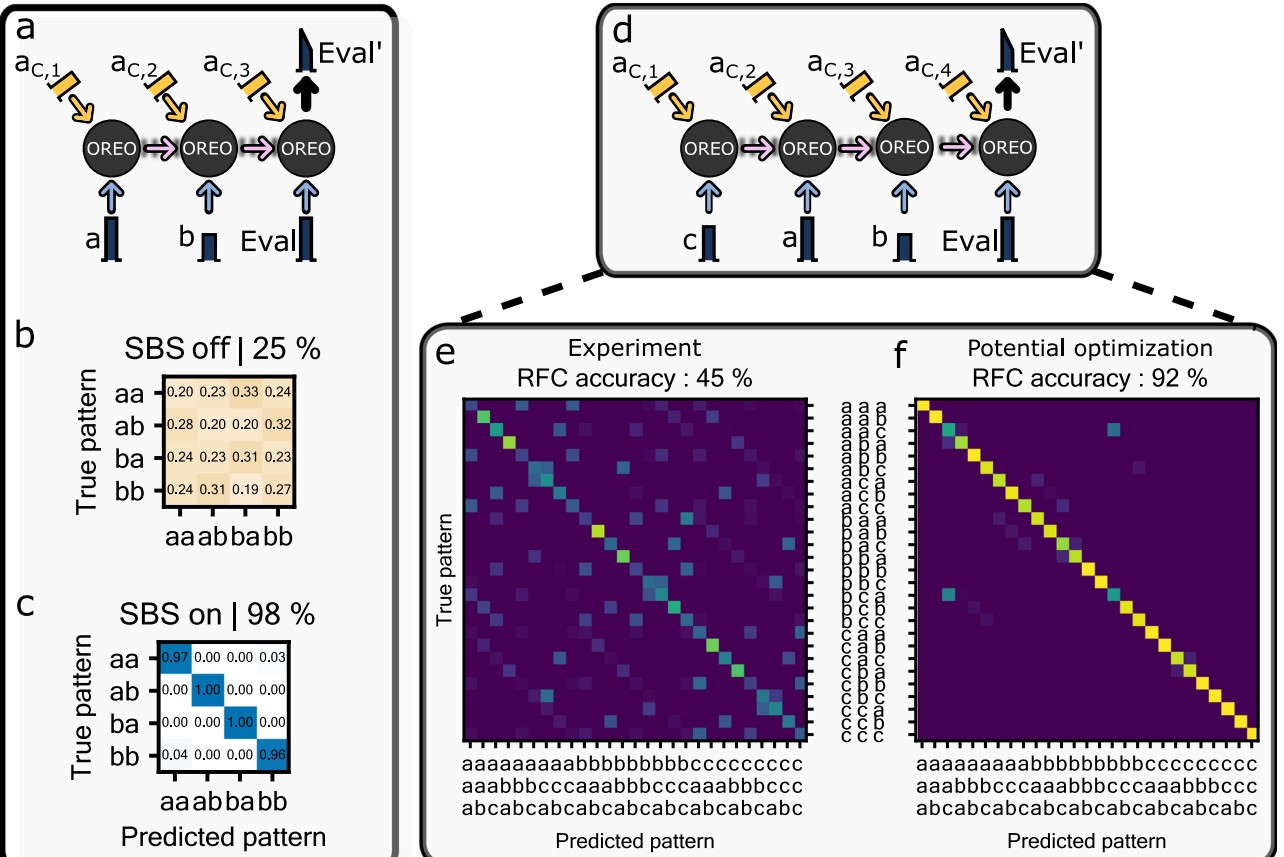

**Fig. 3 | Applying OREO as an acceptor to predict patterns of optical pulses.**
**a** Schematic illustration of how the acoustic link can be used by an optical evaluation pulse (Eval) to predict a pattern from optical pulses, which have been launched into the optical fiber before. The example shows a *ab*-pattern. Note that the output pulses *a′*, and *b′* are not shown. The control pulses $a_{C,i}$ are numbered according to the time sequence of the data pulses as they enter the sample.
**b**, **c** Confusion matrix of a Random Forest classifier (RFC) which is used to classify the experimental data set with and without SBS. The RFC achieves an almost perfect classification rate as soon as OREO provides the recurrent feedback. **d** Schematic

illustration of the three pulse pattern recognition task. This case shows the *cab*-pattern. Note that the output pulses *c′*, *a′*, and *b′* are not shown. **e** Shows the confusion matrix of the RFC, using 30% of the experimentally obtained data set for training. The RFC achieves an accuracy of 45% and outperforms a simple guess by 11-times. The accuracy of the RFC is mostly limited by experimental precision. **f** RFC confusion matrix using simulated data to study OREO's performance with experimental optimization. In this case, we can achieve an accuracy of 92%. The simulation are based on a frequency matched SBS process and is described in more detail in the supplementary material.

on average an accuracy of ≈ 55.9 %. The relative difference to OREO is about 10%P, hence OREO's performance is almost on pair with the digital RNN. However, it should be noted that the digital RNN is trained actively whereas OREO is used as an extreme learning machine. Accordingly, one could improve the accuracy of OREO in the future by training (in-situ) the amplitudes of the control pulses $a_{C,i}$, for instance, using simultaneous perturbation stochastic approximation[46]. Further details on the comparison of OREO with an RNN classifier can be found in the supplementary material.

## Discussion

The acoustic link employed by OREO enables the processing of time-encoded serial information within a PCF. Its capability to control the recurrent interaction all-optically, gives the concept unique features. The adjustable amplitudes of the control pulses allow OREO's behavior to be changed at the single pulse level, offering an all-optical degree of freedom to adjust its recurrent operation. Moreover, we have shown that it offers the possibility to exclude data pulses from the recurrent interaction. As a consequence, a single data pulse can propagate through OREO without experiencing any manipulation. This can be used to implement recurrent dropout as regularization for the RNN.

The coherent nature of the underlying SBS process offers OREO not only to compute amplitude information but also phase

information. Eventually, OREO could compute quadrature amplitude modulated (QAM) data streams.

Higher memory depths could be achieved with three different approaches. Firstly, a higher pulse density could be used to increase the number of operations that could be performed within the intrinsic acoustic lifetime. This could be achieved by decreasing the pulse width and the deadtime between the pulse pairs. For instance, with a pulse width of 100 ps and a deadtime of 100 ps (the minimal deadtime is dictated by the length of the waveguide), one could induce up to 50 recurrent interactions. Secondly, one could increase the acoustic lifetime to realize a deep recurrent link, for instance by using materials with longer acoustic lifetimes or operating at cryogenic temperatures. Thirdly, an optical refreshment of the acoustic waves could lead to an increase in memory depth[47].

Considering the current configuration of the OREO setup, one could realize up to four layers in an OREO-based RNN. In comparison, to best of our knowledge, the maximum number of layers reported in an ONN without transferring information back to the digital domain is so far three[46,48]. A more detailed discussion on the scalability of an OREO-based RNN is provided by the supplementary material.

Because the SBS process does not significantly change the optical control pulses, an optical recycling scheme could be applied to achieve high computational efficiencies. Computational efficiency is

determined by the number of operations (OPS) that OREO can perform with one Joule of power. With an optical recycling scheme this value depends only on the deadtime between the pulse pairs, yielding an efficiency from up to $\approx 11\frac{\text{POPS}}{\text{J}}$; it could potentially increase the computational efficiency of the method described in Reference[22] by three orders of magnitude. A more detailed description of the computational efficiency can be found in the supplementary material. The information bandwidth of an optical signal can be significantly increased by employing different optical frequencies as independent information channels. This has been recently exploited by Sludds et al.[49] to implement an high-performance optical deep learning architecture for edge computing. OREO could be added to this scheme as SBS is highly frequency-selective[50].

This unique feature of the optoacoustic interaction could also be employed together with an optical multi-frequency matrix operator[51–53] to realize an multi-frequency recurrent neural network. Therefore, OREO has to be implemented on-chip. Applications of Brillouin scattering for gyroscopes, photonic memories, and microwave photonic devices haven been experimentally demonstrated in on-chip configurations[26,54–58]. Moreover, different integrated platforms for Brillouin scattering have been discussed extensively[59–61]. Still, a key challenge of using backward Brillouin scattering is the management of the counter propagating data and control pulses. For an in-fiber setup this is usually achieved by using isolators or circulates. The usage of those non-reciprocal devices on-chip is currently more challenging, however, there is extensive study on possible implementations[29–34]. In the context of Brillouin scattering, Liu et al. have demonstrated a complete on-chip backward Brillouin experiment without the need for isolators or circulators[35]. Furthermore, one could also employ a narrowband filter that filterers either the data or the control light. In the following, we eleborate on possible waveguide platforms, the footprint of an SBS active waveguide and the energy conmsuption of on-chip Brillouin process.

Firstly, SBS can be used in a variety of different integratable platforms including Silica, Silicon and Germanium[60]. Backward SBS has also been studied in Silicon Nitride[62,63] and recently reported in Lithium Niobate on Insulator waveguides[64,65]. Moreover, waveguides based on chalcogenide soft glass offer a large Brillouin gain[60,66,67].

Secondly, The footprint of on-chip OREO depends on two properties. First, the pulse width $\tau$ defines the optimal length $L_{opt}$ of the Brillouin active waveguide: $L_{opt} \approx c_0\tau/n_{eff}$ with the speed of light $c_0$ and the effective refractive index of the waveguide. Secondly, the minimal bending radius of the Brillouin active waveguide. This is an important parameter as it allow to route $L_{opt}$ as spiral, which reduces the overall footprint significantly. This reduction of footprint has been used, for instance, to build a 50 cm long Brillouin active spiral in Silicon Nitride[63] and chalcogenide spiral with a total length of 5.8 cm and a footprint below 1 mm$^2$(compare[67]). In addition, bending radii of $\leq 200$ μm have been demonstrated for chalcogenide waveguides[68,69].

Finally, the power consumption of OREO is mainly defined by the optical power of the control pulses $P_c$. In order to estimate the power consumption of an on-chip OREO, we assume to describe the Brillouin dynamics with an effective Brillouin gain $g_{eff} = gL_{eff}P_c - \alpha L$ with the intrinsic Brillouin gain $g$, the optical loss of the waveguide $\alpha$, the entire length of waveguide $L$, the the effective interaction length of two pulses $L_{eff} = (1 - \exp(-\alpha L_{opt}))/\alpha$. By choosing different platforms one effectively changes the values for $g$ and $\alpha$. For instance, the PCF used to demonstrate OREO offers a gain of $g^{PCF} \approx 2.5$ m$^{-1}$ W$^{-1}$ and a loss of $\alpha^{PCF} \approx 2.7$ dB km$^{-1}$ (compare[70]). In comparison, a chalcogenide waveguide offers a gain of $g^{AsS} \approx 500$ m$^{-1}$ W$^{-1}$ with a loss of $\alpha^{AsS} \approx 0.5$ dB cm$^{-1}$ (compare[60]). In this case, one could reduce the required optical power to realize the same $g_{eff}$ by 99% to $P_c^{AsS} \approx 1.27$ mW. Recalling that this is a theoretical improvement and keeping in mind the experimental Brillouin demonstrations on-chip, we would frame this improvement as a long-term goal. Nonetheless, we would like to highlight the

demonstration of a hybrid waveguides[66] offering a gain of $g^{Hybrid} \approx 750$ m$^{-1}$ W$^{-1}$ and the proposal for sub-wavelength waveguides with a gigantic Brillouin gain $g^{sub} > 100000$ m$^{-1}$ W$^{-1}$ (compare[71]). In conclusion, on-chip devices could reduce the power consumption of OREO.

In conclusion, we have demonstrated the first optoacoustic recurrent operator (OREO), which connects the information carried by subsequent optical data pulses. Our work combines for the first time the field of traveling acoustic waves and artificial neural networks and paves the way towards SBS-enhanced computing platforms. This new fusion brings context to optical neural networks, but can also enable much more. Typical building-blocks of a neural network, such as nonlinear activation functions and other types of optoacoustic operators are within reach. Especially, the different time scales of optical and acoustic waves open up a whole new playground for the implementation of a variety of computing architectures.

## Methods
### Experimental setup
To demonstrate OREO, we build the all-fiber setup shown in Supplementary Fig. 1. As a sample, we use a photonic crystal fiber (PCF) with a length of $\approx 40$ cm, an average hole diameter of 1.44 μm, an average core diameter of 1.842 nm, a pitch of 1.756 μm, and $d/\Lambda = 0.82$. A continuous wave laser at 1550 nm is split into the data and control branch via a 50/50-splitter. An IQ-modulator shifts the data signal by $\Omega/2\pi \approx 10.6$ GHz, which is close to the PCF's Brillouin frequency of $\Omega_{PCF}/2\pi \approx 10.45$ GHz. The data signal's spectrum is cleaned with a subsequent narrow bandpass filter and afterwards amplified by an Erbium-doped fiber amplifier (EDFA). An optical intensity modulator driven by an arbitrary waveform generator (AWG) generates the rectangular optical pulses and, thus, imprints the amplitude-encoded information.

A single data pulse is 1 ns long and separated to an adjacent data pulse by a deadtime $dt$. The repetition rate of a pulse sequence is $\approx 1$ MHz. The pulses are guided to the PCF by an optical circulator and, afterwards, measured with a high-speed photodiode and a 16 GHz Oscilloscope. The optical power of the data pulse is about 1 mW. An additional narrow bandpass filter cleans the signal before detection. In the control branch, optical pulses are generated with the same pulse width and repetition rate as the data branch. Afterwards, the pulsed signal is amplified by an EDFA and filtered by a narrow bandpass filter before launched into a high-power EDFA. The amplified signal is filtered by a 1 nm-width bandpass filter and launched with an average power of about 126 mW into the SBS process.

### Training the Random Forest classifier (RFC)
We use a Random Forest classifier (RFC) to classify the area under the curve (AuC) of the output evaluation pulses $Eval'$ (see Fig. 3a, d). We initialize our RFC with the scikit-learn python package v1.1.3[45], setting its maximal depth to 10 and the random state to 2. We found those to be the best configuration to achieve the highest accuracies for both the $ab$- and the $abc$-study. The function sklearn.model_selection.train_test_split splits the initial data sets into the test and training one with a random state of 20. We ensure that that the different patterns are equally distributed to the train and test data set with the stratify-keyword of the train_test_split-function. We fit the RFC to the data set using its default fit-function.

## Data availability
The data that supports findings of this study are available from the corresponding author upon request.

## Code availability
The code that supports findings of this study are available from the corresponding author upon request.

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

## Acknowledgements

The authors thank Florian Marquardt, Christian Wolff, Changlong Zhu, Jesús Humberto Marines Cabello, Toby Bi, and Mark Schöne for fruitful discussions.

## Author contributions

Conceptualization: S.B. and B.S. Methodology: S.B., D.E. and B.S. Investigation: S.B. Visualization: S.B., D.E. and B.S. Funding acquisition: B.S. Project administration: B.S. Supervision: B.S. Writing: S.B., D.E. and B.S.

## Funding

We acknowledge funding from the Max Planck Society through the Independent Max Planck Research Groups scheme, the DFG project STI 792/1-1, and the Studienstiftung des deutschen Volkes. Open Access funding enabled and organized by Projekt DEAL.

## Competing interests

S.B. and B.S. are the authors of a patent filed by the Max-Planck-Gesellschaft zur Förderung der Wissenschaften e.V. about the technique and the features of the optoacoustic recurrent operator. The application numbers are EP23153328.2 for the European Patent Office and PCT/EP2024/051779 for the World Intellectual Property Organization. The remaining author declare no competing interests.
