## [Peer Review File · Nature Communications]

An optoacoustic field-programmable perceptron for recurrent neural networksReviewer #1 (Remarks to the Author):

The authors present an interesting approach to implement recurrence in a photonic optical network by making use of an acoustic feedback path and SBS. The work is interesting, and deserves to be published somewhere. However, for a journal like Nature Communications, the impact is unfortunately too low.

The main issue is that it's not at all clear how this approach would be scaled up from a single neuron to a larger network of neurons. Sticking to the current fibre-based approach is unpractical. But also an integrated approach would have many complications, because of the requirement to have a separate backwards pulse stream for each node, and because of the need for optical isolators, which are far from trivial to integrate.

A second issue is that it's not clear how this approach could be used in more commonly used machine learning tasks, as opposed to a bespoke task of recognising the sequence of 3 pulses. Readout seems to rely on sending in an evaluation pulse after the 3 signal pulses. However, for an input for longer lengths, as needed by general machine learning tasks, the system would presumably not have enough memory to be able to rely on a single evaluation pulse at the end. This could in theory be circumvented by having additional evaluation pulses in the middle, but this would require putting the input 'on hold', which makes the system unsuited for online data processing, i.e. with a continuous unbuffered input stream.

At the very least, results on more demanding tasks (MNIST?) and a realistic approach towards scaling should be presented.

Other remarks:

- The rationale that the authors present for using OREOs as opposed to ring resonators makes little sense. Why would an OREO be more scalable than an integrated ring resonator network? The authors also state that cavities may not be frequency sensitive, but it's trivial to make resonators frequency selective...
- The authors surprisingly claim that the abc task is 7 times as complex as the MNIST task. Surely they don't mean that if they were to try their technique on MNIST, they would expect better performance than for the abc task? First of all, why do they claim that the task has 27^2 possible outcomes, when there are in fact only 27 possible patterns? It does not make any sense to use the size of the confusion matrix here. Second, even though MNIST only has 10 output classes vs 27, the input space is much larger (2^{100} for 100 black and white pixels).
- In discussing Fig. 1, it could be helpful for the reader that the authors stress that the OREO does not implement a simple fixed weight as indicated in the traditional RNN in panel A, but a much more complicated setup where the feedback strengths depend on the input history.
- I would omit the terminology 'acceptor'. It's not widely known and does not really help in explaining the concept.
- Typo: line 72 should probably refer to b_2 rather than b_1
- Typo: line 109: "acoustic wave builds up", rather than "acoustic builds up"?
- Typo: line 151: lower *than*

Reviewer #2 (Remarks to the Author):

This manuscript presents a novel concept of using backward-propagating Stimulated Brillouin Scattering (SBS) for generating acoustic waves in waveguiding media by which it offers optically controlled short-term memory in all-photonic Recurrent Neural Network (RNN). The idea that is elaborated is very interesting and looks promising. Since SBS is a well-understood nonlinear process, it gives additional credibility to the OREO (optoacoustic recurrent operator) and the authors take a

very methodic approach in describing its mechanics in Supplementary material (Ch. S6). Overall, the manuscript meets the guidelines in terms of content: there is background/theory section, experiment aiming to demonstrate the SBS effect in OREO and a simple use-case of pattern recognition. However, my overall impression is that the manuscript is not at the maturity level required for publication in Nat Comm, at least in the current form. Following a revision, I would be open to reconsider my opinion.

Questions/comments for the authors:

1. Description of the experimental results in pulse amplitude sweeps could be improved. It is unclear how single pulse interaction (SPI) is defined and normalized. What insights can we get from comparing the sweep results to SPI? How should SPI value of ~ 0.53 in Fig. 2 be understood? Overall, better description is needed for the process of normalization and acoustic wave interference and its implications since it is quite challenging to interpret the results.
2. What kind of pulses have been used in experiments? Does the pulse shape play any role in OREO's response? Is there any preferable shape? How is the deadtime defined – as a repetition rate or as an empty slot between the pulses? If the 2nd definition is used, which criterion is used for pulse start/stop in case of non-rectangular pulses (e.g., Gaussian)?
3. The authors mention that OREO is used as an acceptor (lines 92, 220), which, I assume, means as a perceptron in the last layer of the RNN? Was there any layer preceding the last one? If so, how many? What was the size of each layer? Were these implemented in software? Overall, more details are required to understand the results shown for pattern recognition.
4. In pattern recognition experiment, it would be much more interesting to compare the results achieved by OREO-RNN to the results of other RNNs (photonic/electronic), maybe by using a dataset for which accuracy results have been published. Comparing it to the network with recurrent behavior turned off does not bring much value.
5. The authors encode letters to pulse amplitudes during pattern recognition. What is the data-pulse resolution that OREO can resolve, or, in other words, how many unique symbols you could potentially use in a sequence?
6. In lines 85-88 of the introduction, the authors claim that OREO could soon be implemented as a fully integrated design. It would be very beneficial to elaborate this claim either in the main manuscript or supplementary document and provide some context. Which kind of waveguiding elements would have to be used, how would this impact control pulse power and deadtime, what would be the power consumption/dissipation, what would be OREO node footprint etc., so that the platform's practicability can be better judged.
7. In the methods section, the authors mention the optical power per data pulse of $\sim 1\text{mW}$. How does this influence the number of layers that could be used in NN? Some estimate of scalability of the network would be appreciated.
8. Optical power per control pulse of $> 100\text{ mW}$ for photonic crystal fiber (PCF) is extremely high and will drive up power consumption even under the assumption that control pulses can be recycled. Are there any other waveguiding candidates that could operate with lower power control pulses? What are the estimates for control pulse power in case of integrated solutions?
9. The authors claim that OREO might reach the accuracy of 97% (Line 253 and Supplementary Fig. 7). How does this compare with available prediction models?
10. In Supplementary material, Fig S3, the authors say that for B-E only acoustic wave b1 is observed and for F-I only b2. In the description of the procedure in section S4 it is said that the control pulse

that is not swept is being kept constant. How does this eliminate the influence of the other acoustic wave? I would assume that aD1-aC1 interaction (yielding b1) will be there while aC2 is being swept. Could the authors clarify?

Minor comments/questions:

1. Figure 2 in main manuscript: make SPI line more visible, it is very challenging to spot it.
2. Figure S2 in Supplementary document could be complemented by correlation plots followed by comparison of the correlation coefficients w/out and w/ SBS (both with detuned and tuned delay).
3. Is ROPS in S3 a typo or it somehow differs from OPS?

Reviewer #3 (Remarks to the Author):

In this work, the authors employ the theory of stimulated Brillouin scattering (SBS) to demonstrate optoacoustic recurrent operators (OREOs) which are very crucial for the short-term memory requirement of optical recurrent neural networks. In contrast to the rapid progress in optical deep neural networks utilizing either diffractive optics or photonic interferometric mesh in recent years, the optical analogy of recurrent neural network has not been well established. In this regard, the result of this work is very intriguing and promising for optical neuromorphic processing especially of temporal data.

As being not familiar with SBS, the only concern for me is that the mechanism of SBS and its operation principle described in experimental results are quite complicated and not easy to catch up, while such in-depth understanding is essential to assess the importance of the manuscript. For instance, the mechanism of SBS as a link between optical pulses is not clearly seen through Fig. 1 and the related description. When $a_{D,1}$ encounters $a_{C,1}$, they generate b_1 without any prior existence of acoustic waves. After then, does the subsequent SBS between $a_{D,2}$ and $a_{C,2}$ requires b_1 to generate b_2 ? It would be better to clarify whether any pair of $a_{D,i}$ and $a_{C,i}$ generates b_i regardless of prior b_j for $j < i$, and so what the difference is between SBS with and without prior b 's. While I could find the hints in the subsequent Fig. 2 (does this comparison correspond to line 151?) and supplementary information S6, in my opinion, it should be commented earlier.

Overall, I would recommend the acceptance after the above clarifications, considering the general readership of Nature Communications. The followings are some minor points.

1. Typo in Fig. 1A: the final weight matrix seems to be W^{N-1} .
2. Two sentences are very confusing: "b1 weakens..." in line 152 and "b1 enhances..." in line 157.
3. Is there any possibility of interactions for data/control pulses $a_{D,i}$ and $a_{C,j}$ for different i and j at different position?
4. The details of the RFC and the way how it is trained could be described in Methods section.

Reply to Reviewer 1

Comment 1: *The authors present an interesting approach to implement recurrence in a photonic optical network by making use of an acoustic feedback path and SBS. The work is interesting, and deserves to be published somewhere. However, for a journal like Nature Communications, the impact is unfortunately too low.*

Response 1: We thank the reviewer for taking the time to evaluate our experimental demonstration of the first optoacoustic recurrent operator. We appreciate the challenge provide by the reviewer to improve our manuscript and to stress its impact.

In particular with the support of reviewer 3, pointing out that *“optical analogy of recurrent neural network has not been well established. In this regard, the result of this work is very intriguing and promising for optical neuromorphic processing especially of temporal data”*, we revised the manuscript to convince reviewer 1. Considering that Nature Communication has published interesting work on building blocks of optical neural networks before, including theoretical frameworks such [1], we firmly believe that the following reply will convince the reviewer and the editor that our work meets the requirements of Nature Communication.

Comment 2: *“The main issue is that it's not at all clear how this approach would be scaled up from a single neuron to a larger network of neurons. Sticking to the current fibre-based approach is unpractical. But also an integrated approach would have many complications, because of the requirement to have a separate backwards pulse stream for each node, and because of the need for optical isolators, which are far from trivial to integrate.”*

Response 2: We thank the reviewer for the critical reflection of our architecture and the need for optical isolators. We agree that implementing OREO on a chip is of great importance. Indeed, applications of Brillouin scattering for gyroscopes, photonic memories, and microwave photonic devices haven been experimentally demonstrated in on-chip configurations.

Our manuscript serves as proof of principal study on which new possibilities stimulated Brillouin scattering can offer for the field of photonic machine learning and optical fibers are a great platform to do so.

In general the capability for integrated Brillouin applications have been studied extensively[2], [3]. Still, we agree that a key challenge of using backward Brillouin scattering is the management of the counter propagating data and control pulses. There is extensive study on possible implementations of on-chip isolators and circulators [31–36]. In the context of Brillouin scattering, Liu et al. have demonstrated a complete on-chip backward Brillouin experiment without the need for isolators or circulators[4]. Furthermore, one could also employ a narrowband filter that filterers either the data or the control light.

Changes to the manuscript:

We added a detailed description on “on-chip OREO” to the discussion section of the main text.

[Main text, p. 5]

Previously:

“In particular, with the announcement of the first on-chip EDFA a fully integrated design is seemingly close.”

Updated:

“With the announcement of the first on-chip EDFA a fully integrated design is in reach. While recognizing the outstanding efforts of the scientific community to implement non-reciprocal devices on-chip, we would like to highlight the work [35] which implements an on-chip circulator free SBS experiment. More details about how OREO could be integrated on-chip can be found in the discussion.”

Comment 3: *“A second issue is that it's not clear how this approach could be used in more commonly used machine learning tasks, as opposed to a bespoke task of recognising the sequence of 3 pulses. Readout seems to rely on sending in an evaluation pulse after the 3 signal pulses. However, for an input for longer lengths, as needed by general machine learning tasks, the system would presumably not have enough memory to be able to rely on a single evaluation pulse at the end. This could in theory be circumvented by having additional evaluation pulses in the middle, but this would require putting the input 'on hold', which makes the system unsuited for online data processing, i.e. with a continuous unbuffered input stream.”*

Response 3: We would like to thank the reviewer for pointing out the current depth limitation of OREO. The current manuscript shows a four-pulse interaction which is indeed a limiting factor at the moment. While addressing a more general machine learning task, we would like to point out that the first digital recurrent neural networks[5], were only capable of making connections of the same order of magnitude as the current version of OREO. And even nowadays, simple RNNs cannot bridge more than five to ten connections due to vanishing error problems[6]. Hence, our approach is almost on par with standard RNNs.

In order to show future research directions which, increase the number of recurrent connections, we mention in the manuscript's conclusion two possibilities:

Firstly, the usage of shorter pulses can increase the number of connections. For instance, with 500ps wide pulses, one could achieve at least eight connections.

Secondly, one can operate OREO in platforms or environments with a higher phonon lifetime. For instance, one can increase the intrinsic phonon lifetime[7]. We would like to point out that since submission, we have used the approach of [7] to demonstrated a link of eight pulses.

In conclusion, the number of connections realized by the current version of OREO is almost on pair with the first recurrent neural networks published about 25 years ago. Hence, we would hope that our proof of principle of a first photonic recurrent operation with pulse-by-pulse control is a similar inspiration as those early-on RNN, leading to much better performance in the future.

Comment 4: *“It the very least, results on more demanding tasks (MNIST?) and a realistic approach towards scaling should be presented.”*

Response 4:

We would like to thank the reviewer for mentioning the MNIST image classification as a potential task for OREO. Sequential MNIST classification task is indeed a demanding task, where even standard digital RNNs fail to work[8]. Hence, it is not an appropriate task for our proof-of-principle and is, indeed, interesting for future work.

However, our work follows a usual approach of the digital RNN community, using synthetic tasks to demonstrate essential functionalities of RNNs[9], [10]. The functionality that we are demonstrating

is to transform a in time multi-dimensional input space in a single dimension, the core functionality of an acceptor and a functionality needed for language processing[11].

Whereas the Elman paper[12], being one of the first paper on language processing, used six different letters for a sequence prediction, OREO is used to classify twenty-seven different classes of pulse patterns. Pointing to an argument of Elman[12], such patterns could also be seen as a representation of speech sounds although they are not sampled from real speech.

Hence, the performed task could be seen as a synthetic task to demonstrate the fundamental language processing capabilities of OREO.

In conclusion, we are convinced that the abc-task serves as a synthetic task to demonstrate the core strength of our proof-of-principle which could inspire future work.

Furthermore, we would like to address the question about scalability in the following: In order to do so, we must distinguish between different types of architecture of the underlying optical neural network (ONN) which could use OREO:

Feed-forward (FF) architecture: This type of ONN feeds the output of layer n directly into layer n+1 without amplifying the light between the layers. For instance, the FF-arch. is used by[13].

Supply light (SL) architecture: This type of ONN uses the output of layer n to manipulate a supply light channel which is feed into layer n + 1. The power in the supply light channel is constant for all layers. For example. the SL-arch. is used by[14].

Optical-Digital-Optical (ODO) architecture: This type of ONN converts the output of layer n into the digital domain. The digital information is then transferred back into the optical domain for layer n+1, for instance, via Mach-Zehnder modulators or attenuators. Such an approach is used by[15], [16].

In addition, one has to distinguish if the signal is detected with a direct or homo-/heterodyne detection scheme. In the latter one, the power of the local oscillator P_{LO} has to be taken with into account. The minimal input power that can be detected is noted as P_{min} and depends on the detector's Noise-Equivalent Power (NEP) and its bandwidth.

The maximum numbers layers N for a given input power P_{in} of an OREO-based RNN, where each layer has an insertion loss α_{layer} , is summarized in

Table 1 Scalability of an OREO-based RNN. All powers in [dBm] and the insertion loss in [dB]. We assume the minimal signal power detectable with homo-/heterodyne detection to be $P_{min}^{HD}(P_{LO}) \approx (P_{min})^2/(2 P_{LO})$ (compare [17]).

	FF-arch	SL-arch. and ODO-arch.
Direct Detection	$N = \frac{(P_{in} - P_{min})}{\alpha_{layer}}$	Independent of P_{in} as long as $P_{in} - \alpha_{layer} > P_{min}$
Homo-/ Heterodyne detection	$N \propto \frac{(P_{in} - P_{min}^{HD}(P_{LO}))}{\alpha_{layer}}$	Independent of P_{in} as long as $P_{in} - \alpha_{layer} > P_{min}^{HD}(P_{LO})$

The current design of OREO has an intrinsic loss of 2 dB due to the used circulators plus 3 dB of the pulse depletion. Note that the latter value depends on the exact configuration of OREO (compare Figure 2 of the main text), and we assume that on average it represents the single pulse interaction (SPI) value. A complete layer of an RNN is formed if OREO is combined with a matrix multiplication and a nonlinear activation function, which we assume to have a loss of 1.32 dB as reported by [13]. Hence, total insertion loss per layer $\alpha_{layer} \approx 6.32$ dB. With the minimal detection power $P_{min} \approx -24$ dBm of the used photodetector and the initial input power of $P_{in} \approx 0$ dBm, one could realize about four layers. In comparison, the maximum number of layers reported in an ONN without transferring information back to the digital domain is so far three [13], [14].

In order to increase the number of layers in the future of an OREO-based RNN, one could use a lower SPI-depletion value. For instance, one could convert only 10 % of the data pulse into the acoustic domain, yielding a loss of ≈ 0.5 dB. This would yield a maximum number of ≈ 6 layers that could be realized. Another two layers could be added by replacing the current Newport 12 GHz photodetectors with 10 GHz photodetectors manufactured by Thorlabs. Eventually, one can exchange direct detection with homo-/heterodyne detection which has been already used in [13]. In this case, one could push the maximum number of layers to approximate 16, considering a local oscillator power of $P_{LO} = 1$ mW.

Changes to the manuscript:

1. *[Supplementary information] Added section S4, discussing the scalability of OREO.*
2. *[Main text, p. 15] Added the following text:*
“Considering the current configuration of the OREO setup, one could realize up to four layers in an OREO-based RNN. In comparison, to best of our knowledge, the maximum number of layers reported in an ONN without transferring information back to the digital domain is so far three [13], [14]. A more detailed discussion on the scalability of an OREO-based RNN is provided by the supplementary information.”

Comment 5: *The rationale that the authors present for using OREOs as opposed to ring resonators makes little sense. Why would an OREO be more scalable than an integrated ring resonator network?*

Response 5: We do not intend to diminish the impact of the ring resonator network approach to the neural network community. These works are absolutely impressive and inspiring to us. We might not have found the right formulation for the comparison. What we intended to highlight are the well-known difficulties arising from the additional manufacturing process which led to variations in the properties. The properties of micro ring resonators such as the quality factor are vulnerable to those changes [18]. In particular, the free spectral range (FSR) of a ring resonator is a very sensitive parameter for the optimization of a recurrent operation. The FSR is an important parameter to synchronize the recurrent response of the resonator with the deadtimes of an input pulse sequence. Hence it must be adjusted to match different deadtimes. However, a change in FSR also changes the resonance of the resonator and therefore its coupling strength and eventually the degree of recurrent feedback. This correlation limits the scalability of a ring resonator network as it limits the network to specific configurations. We would like to point out further, that OREO uses the intrinsic properties of the waveguide and does not add an additional footprint.

Changes to the manuscript:

[Main text, p.3]

Previously:

“Firstly, the usage of an artificial cavity, e.g. a ring resonator, can limit the scalability of those networks.”

Updated:

“Firstly, the usage of an artificial cavity, e.g., a ring resonator, requires additional tuning of the individual rings due to manufacturing depended properties such as the free spectral range (FSR). This requires additional compensation routines in order to match the FSR with the chosen temporal dynamic, wavelength and coupling.”

Comment 6: *“The authors also state that cavities may not be frequency sensitive, but it's trivial to make resonators frequency selective... - The authors surprisingly claim that the abc task is 7 times as complex as the MNIST task. Surely they don't mean that if they were to try their technique on MNIST, they would expect better performance than for the abc task? First of all, why do they claim that the task has 27^2 possible outcomes, when there are in fact only 27 possible patterns? It does not make any sense to use the size of the confusion matrix here. Second, even though MNIST only has 10 output classes vs 27, the input space is much larger (2^{100} for 100 black and white pixels).”*

Response 6: We would like to thank the reviewer for pointing out that our wording while stating possible disadvantages of s free-space cavities, delay systems, and micro ring resonators was misleading. In the manuscript we refer to “cavities” are not being frequency selective, when we meant delay line systems and free-space cavities and not to mirroring resonators. We agree with the reviewer that micro ring resonators are indeed frequency selective, a property which have been frequently used for photonic machine learning. We will adapt the wording to clarify the distinguishing.

Furthermore, we would like to thank the reviewer for pointing out the simplified comparison of the abc-classification and MNIST dataset classification task. The input space argument presents a valid counterpoint, suggesting that the complexity assessment should consider not only the number of output classes but also the dimensionality of the input data.

Changes to the manuscript:

1. [Main text, p. 3]

Previously:

“Secondly, the cavity may not be frequency sensitive, preventing them from being applied for resource-efficient multi-frequency data processing.”

Updated:

“Secondly, the free-space cavities and delay systems may not be frequency sensitive, preventing them from being applied for resource-efficient multi-frequency data processing.”

2. [Main text, p. 12]

Removed:

“Note that the classification task of the abc-case has 729 possible classification outcomes, and is 45 times more complex as the ab-case. In general, it is seven times more complex as an image classification task based on the MNIST dataset with 10×10 degrees.

Comment 7: “In discussing Fig. 1, it could be helpful for the reader that the authors stress that the OREO does not implement a simple fixed weight as indicated in the traditional RNN in panel A, but a much more complicated setup where the feedback strengths depend on the input history.”

Response 7: We thank the reviewer for pointing out that Fig 1 A needs more clarification about the recurrent task. However, we are not sure if we have correctly understood the reviewer's point of additional complexity and would appreciate additional clarification if the following explanation does not clear up a misunderstanding.

In our view, a traditional RNN scheme consist of a linear operation applied to the input, which is in Figure 1 represented by matrix “W” and the recurrent feedback.

As for a traditional RNN, OREO’s recurrent feedback depends on the input history. The amount of input that is used for the recurrent feedback is defined by the corresponding control pulses which is shown in Figure 1 B. Hence, from our perspective the complexity, a recurrent neuron formed with OREO has the similar complexity as a simple digital recurrent neuron.

Reply to Reviewer 2

Comment 1: *This manuscript presents a novel concept of using backward-propagating Stimulated Brillouin Scattering (SBS) for generating acoustic waves in waveguiding media by which it offers optically controlled short-term memory in all-photon Recurrent Neural Network (RNN). The idea that is elaborated is very interesting and looks promising. Since SBS is a well-understood nonlinear process, it gives additional credibility to the OREO (optoacoustic recurrent operator) and the authors take a very methodic approach in describing its mechanics in Supplementary material (Ch. S6). Overall, the manuscript meets the guidelines in terms of content: there is background/theory section, experiment aiming to demonstrate the SBS effect in OREO and a simple use-case of pattern recognition. However, my overall impression is that the manuscript is not at the maturity level required for publication in Nat Comm, at least in the current form. Following a revision, I would be open to reconsider my opinion.*

Response 1: We thank the reviewer for the in principle positive impression of our work, pointing out the novelty and credibility of our work.

Comment 2: *“Description of the experimental results in pulse amplitude sweeps could be improved. It is unclear how single pulse interaction (SPI) is defined and normalized. What insights can we get from comparing the sweep results to SPI? How should SPI value of ~ 0.53 in Fig. 2 be understood? Overall, better description is needed for the process of normalization and acoustic wave interference and its implications since it is quite challenging to interpret the results.”*

Response 2: We would like to thank the reviewer for pointing out the need for additional information. We will clarify our description.

Single pulse interaction (SPI) refers to the degree of depletion of a single data-control interaction without the influence of an acoustic wave. It showcases that about 47% of the data pulse is converted into the acoustic domain. If one employs weaker optical control pulses the value for SPI will decrease as less data is written into the acoustic. As it determines the strength of the Brillouin process it could be used in future publications on the optoacoustic recurrent operator as reference.

We normalize the AuC of the output pulses $a'_{\{D, i\}}$ with the AuC of an input data pulse $a_{\{D, i\}}$ with an amplitude scale of 1, representing the highest input value. In order to measure the reference, we launch a data pulse sequence into OREO without counter-propagating control pulses.

Changes to the manuscript:

1. [Main text, p. 7]

Previously:

“For each amplitude step, we measure the pulse’s area under curve (AuC) of the output pulses $a'_{\{D, i\}}$. An AuC-measurement of $a_{\{D, i\}}$ without control pulses serves as reference. In total, three different acoustic links occur from this experimental configuration, namely, $a_{\{D, 1\}} \rightarrow a'_{\{D, 2\}}$, $a_{\{D, 1\}} \rightarrow a'_{\{D, 3\}}$, and $a_{\{D, 2\}} \rightarrow a'_{\{D, 3\}}$ (see Figure A).”

Updated:

“In the subsequent analysis, we normalize the AuC of the output pulses $a'_{\{D, i\}}$ with the AuC of an input data pulse $a_{\{D, i\}}$ with an amplitude scale of 1, representing the highest input value. In order to measure the reference, we launch a data pulse sequence into OREO without counter-propagating control pulses.

2. [Main text, p. 8]

Previously:

“Because the degree of depletion is lower as for a single pulse interaction (SPI), e.g., $a_{D,1} \rightarrow a'_{D,3}$, we conclude that the acoustic wave b_1 weakens the SBS process of $a_{D,2} \leftrightarrow a_{C,2}$.”

Updated:

“In order to have a reference for the dynamic of the Brillouin process, we use the interaction of a single data-control pulse pair. The data pulse a_D gets depleted by the SBS process in this single pulse interaction (SPI). The degree of depletion depends on the power carried by the control pulse[19]. For OREO, we deplete about 47 % of the data pulse.”

3. [Main text, p. 8] Added the following text block:

“For example, if the interaction $a_{D,i} \leftrightarrow a_{C,i}$ has created the acoustic wave b_i with a specific phase (see Equation (2)), then a second SBS process $j>i$ would generate dynamically b_j on top of b_i . Both b_i and b_j have a different phase which lead to acoustic interference [34]. As Brillouin scattering describes a dynamic coupling between photons and phonons, the acoustic interference effects the overall "stimulated" dynamic. Hence, if b_i and b_j interfere destructively, the “stimulated” dynamic is slowed down and, therefore, the depletion of the data pulse $a_{\{D, j\}}$ lower. However, if b_i and b_j add up via constructive interference, then the stimulation is accelerated, increasing the depletion of $a_{\{D, j\}}$.”

Comment 3: “What kind of pulses have been used in experiments? Does the pulse shape play any role in OREO’s response? Is there any preferable shape? How is the deadtime defined – as a repetition rate or as an empty slot between the pulses? If the 2nd definition is used, which criterion is used for pulse start/stop in case of non-rectangular pulses (e.g., Gaussian)?”

Response 3: We would like to thank the reviewer for pointing out that we did not clarify clearly enough the used pulse parameters. While we are discussing the pulse width and the deadtime in the “Methods”-section of the main text, we are only showing indirectly the pulse shape (rectangular) in the Figures 1, 2, 3 and 4. We will name it directly in the manuscript.

In addition, the reviewer understood the deadtime correctly as “empty slot”, which means an off-time between two Brillouin processes. This is a criterion which is independent of the pulse shape. In the case of Gaussian pulses, one has to obey the fact that the width σ of the pulse does not defines the end of the pulse. As a rule of thumb, one could use here 3σ which ensures that there is a neglectable overlap between the pulses.

For a single-frequency and frequency-matched Brillouin process, the pulse shape has little impact. In this case, the pulse area is the essential parameter which influences OREO’s response. In the frequency-mismatched case the spectral shape of the pulses influences the Brillouin process[20], [21]. In addition, if one wants to employ multiple frequency channels to run OREO in parallel, the pulse shape only has an impact on how close one can position two frequency channels[20], [21].

Changes to the manuscript:

1. [Main text, p.6]

Previously:

“Here we launch several consecutive optical input data pulses $a_{D,i}$ and strong counter-propagating optical control pulses $a_{C,i}$ into a PCF.”

Updated:

“Here we launch several consecutive rectangular optical input data pulses $a_{D,i}$ and strong counter-propagating optical control pulses $a_{C,i}$ into a PCF.”

2. [Main text, p.18-19]

Previously:

“An optical intensity modulator driven by an arbitrary waveform generator (AWG) generates the optical pulses and, thus, imprints the amplitude-encoded information.”

Updated:

“An optical intensity modulator driven by an arbitrary waveform generator (AWG) generates the rectangular optical pulses and, thus, imprints the amplitude-encoded information.”

Comment 4: “The authors mention that OREO is used as an acceptor (lines 92, 220), which, I assume, means as a perceptron in the last layer of the RNN? Was there any layer preceding the last one? If so, how many? What was the size of each layer? Were these implemented in software? Overall, more details are required to understand the results shown for pattern recognition.”

Response 4: We would like to thank the reviewer for pointing out the need for additional clarification about OREO functioning as an acceptor. We would like to stress that we are experimentally demonstrating those capabilities in the paper and are not implementing a digital-twin. Only the final classification of the output evaluation pulse $Eval'$ is performed by a digital Random Forest classifier (RFC).

The general information flow of an acceptor is illustrated in Figure 1 Information flow of an acceptor Where the sequential input information x_i with $i = 1,2,3,4$ is transformed into the output state y_4 , which is then used to classify the input sequence x_i (see Reference [22]). Ideally, the output state is unique to a corresponding input state, which allows a subsequent classifier to distinguish the different input states with high precision. In the literature one finds also the term RNN recognizer for an RNN acceptor[23].

Figure 1 Information flow of an acceptor with the input x_i , the state h_i , and the output y_i .

A similar graph for OREO working as an acceptor is shown in Figure 2 OREO used to create an acceptor. Hence, our scheme works as single recurrent perceptron, which performs the abc-recognition task with a digital Random Forest classifier (RFC).

Figure 2 OREO used to create an acceptor. The output data pulses $a'_{D,i}$ are not shown.

Changes to the manuscript:

1. [Main text, Figure 3]

Replaced the schematic illustration of the abc-study to highlight the information flow. We also updated the caption of the figure.

2. [Main text, p.11]

Previously:

“In the following section, we employ OREO as an acceptor to recognize any pattern that can be created with two different data pulses a and b: aa, ab, ba & bb, where the b-pulse is half the amplitude of the a-pulse.”

Updated:

“In the following section, we experimentally employ OREO as an acceptor to recognize any pattern that can be created with two different data pulses a and b: aa, ab, ba & bb, where the b-pulse is half the amplitude of the a-pulse.”

3. [Main text, p.12]

Previously:

“As a measure of OREO’s performance, we launch an evaluation pulse pair into the optical fiber and use the AuC of an output evaluation pulse (Eval) (see Figure 3 A). In total, we check all patterns 250-times in a random order and classify the resulting data set (70%training, 30%testing) with a Random Forest classifier (RFC) implemented in sklearn-package.”

Updated:

“Figure 3 A shows schematically the information flow of the acceptor task. The individual pulses of the pattern ab are launched into OREO each after 2.5 ns with a corresponding control pulse $a_{C,i}$. The information of the pattern is captured by the acoustic wave and flows in time connecting the different SBS processes. Eventually, the sequential information captured by OREO is evaluated with a third optical

evaluation pulse (Eval). In this way, we map the sequential information onto a single optical pulse which can be used then for classification, realizing experimentally a photonic acceptor. We perform the final classification by measuring the output evaluation pulse (Eval'), extracting its AuC in post-processing and then feeding this value into a digital RandomForest classifier (RFC). In total, we check all patterns 250-times in a random order and classify the resulting experimentally obtained data set (70 % training, 30 %testing) with the RFC implemented in sklearn-package."

4. [Main text, p.12]

Previously:

"We increased the sample size n per pattern from 250 to 500 in order to decrease statistical errors."

Updated

"We increased the sample size n per pattern from 250 to 500 measurements in order to decrease statistical errors."

Comment 5: *"In pattern recognition experiment, it would be much more interesting to compare the results achieved by OREO-RNN to the results of other RNNs (photonic/electronic), maybe by using a dataset for which accuracy results have been published. Comparing it to the network with recurrent behavior turned off does not bring much value."*

Response 5: We would like to thank the reviewer for pointing out the need for a reference. As suggested by the reviewer, we include in the revised manuscript a comparison of the OREO with a digital RNN. In the additional study, we found that RNN's performance on the proposed pattern recognition task is almost on par with OREO.

Changes to the manuscript:

1. We added section S9 to the supplementary material to elaborate on the comparison of OREO with an digital RNN.
2. [Main text, p. 13] Added the following text block:

"This classifier achieves in the same task on average an accuracy of 55.9 %. The relative difference to OREO is about 10 %P, hence OREO's performance is almost on pair with the digital RNN. However, it should be noted that the digital RNN is trained actively whereas OREO is used as an extreme learning machine. Accordingly, one could improve the accuracy of OREO in the future by training (in-situ) the amplitudes of the control pulses $a_{\{C, i\}}$, for instance, using stochastic gradient descent[13]"

Comment 6: *"The authors encode letters to pulse amplitudes during pattern recognition. What is the data-pulse resolution that OREO can resolve, or, in other words, how many unique symbols you could potentially use in a sequence?"*

Response 6: We would like to thank the reviewer for pointing out the number of symbols one can potentially process with OREO. Generally speaking, the SBS process is sensitive to finite separated data inputs and OREO can be used in general for analog processing scheme. In the amplitude sweep we used 9 steps which are distinguishable leading to a 3bit precision, however the dynamics in the plots suggest that 4 or 5 bits can also be resolved. For instance, in the context of an optoacoustic memory

which stores a certain amplitude information for a certain time period a linear relation between input and output amplitude was demonstrated[24], which stresses the point that multiple bit of information can be processed. Eventually, the number of symbols is not limited by the physics but by the encoding and detecting devices.

Changes to the manuscript:

[Added to the main text, p. 9]

“Being able to distinguish nine different amplitude levels, we conclude that OREO can resolve at least 3-bit information. In general, SBS is sensitive to finite separated amplitude levels, hence higher number of unique symbols can be encoded. The fundamental limitation is the data encoding and detection.”

Comment 7:” In lines 85-88 of the introduction, the authors claim that OREO could soon be implemented as a fully integrated design. It would be very beneficial to elaborate this claim either in the main manuscript or supplementary document and provide some context. Which kind of waveguiding elements would have to be used, how would this impact control pulse power and deadtime, what would be the power consumption/dissipation, what would be OREO node footprint etc., so that the platform’s practicability can be better judged.”

Response 7: We would like to thank the reviewer for stressing the need for additional elaboration on how to use OREO on-chip. We will provide context about how to use OREO on-chip in the discussion section of the main text. SBS has been discussed extensively for on-chip applications[2], [3], and many different waveguide platforms, have been used, for instance chalcogenide soft glass As_2S_3 , Silica, Silicon, Germanium, Silicon Nitride and Lithium niobate on insulator[25].

What would be the impact on the mentioned parameters?

Deadtime:

The Deadtime depends on the waveguide length Hence, an integrated platform would not lead to on-chip specific change on the deadtime.

Power consumption:

The power consumption of OREO is mainly related to the control pulse power $P_{control}$ which is chosen to induce the SBS process., The underlying effective efficiency of a Brillouin process g_{eff} can be approximated as

$$g_{eff} = g L_{int} P_{control} - \alpha L$$

with the Brillouin gain g_{SBS} , the effective interaction length of the two pulses L_{int} , the optical loss α and the length of the Brillouin-active waveguide $L \geq L_{int}$.

By choosing different integrated platforms one effectively changes the values for g and α . For instance, the used PCF has a gain of $g^{PCF} \approx 2.5 m^{-1}W^{-1}$ and a loss of $\alpha^{PCF} \approx 2.7 dB/km$ [26]. In comparison, an on-chip chalcogenide waveguide can have a gain of $g^{AsS} \approx 500 m^{-1}W^{-1}$ with a loss $\alpha^{AsS} \approx 0.5 dB/cm$ [2]. In this case one can reduce the required optical power to generate the same g_{eff} by 99% to 1.27 mW. Recalling that this is a theoretical improvement and keeping in mind the experimental Brillouin demonstrations on-chip, we would frame this improvement as a long-term goal. Nonetheless, we would like to highlight the demonstration on hybrid waveguides[27] with a gain of $g^{hybrid} = 750 m^{-1}W^{-1}$ and the proposals[28] for sub-wavelength waveguides with gigantic Brillouin gain values $g > 10000 m^{-1}W^{-1}$.

The node footprint:

The footprint of on-chip OREO depends on two properties. First, the pulse width τ defines the optimal length L_{opt} of the Brillouin active waveguide: $L_{opt} \approx c_0\tau / n_{eff}$ with the speed of light c_0 and the effective refractive index of the waveguide. Secondly, the minimal bending radius of the Brillouin active waveguide. This is an important parameter as to route L_{opt} as spiral, which reduces the overall footprint significantly. This reduction of footprint has been used, for instance, Silicon Nitride to build a 50 cm long spiral and a chalcogenide spiral with a total length of 5.8 cm and a footprint below 1 mm. In addition, bending radii of $\leq 200 \mu\text{m}$ have been demonstrated for chalcogenide waveguides.

Changes to the manuscript:

1. We added a detailed description on “on-chip OREO” to the discussion section of the main text.
2. [Main text, p. 5]

Previously:

“In particular, with the announcement of the first on-chip EDFA a fully integrated design is seemingly close.”

Updated:

“With the announcement of the first on-chip EDFA a fully integrated design is in reach. While recognizing the outstanding efforts of the scientific community to implement non-reciprocal devices on-chip, we would like to highlight the work which implements an on-chip circulator free SBS experiment. More details about how OREO could be integrated on-chip can be found in the discussion.”

Comment 8: *“In the methods section, the authors mention the optical power per data pulse of $\sim 1\text{mW}$. How does this influence the number of layers that could be used in NN? Some estimate of scalability of the network would be appreciated.”*

Response 8: In order to answer the question, we have to distinguish different types of architectures of the underlying optical neural network (ONN):

Feed-forward (FF) architecture:

This type of ONN feeds the output of layer n directly into layer $n+1$ without amplifying the light between the layers. For instance, the FF-arch. is used by [13].

Supply light (SL) architecture:

This type of ONN uses the output of layer n to manipulate a supply light channel which is feed into layer $n + 1$. The power in the supply light channel is constant for all layers. For example, the SL-arch. is used by [14].

Optical-Digital-Optical (ODO) architecture:

This type of ONN converts the output of layer n into the digital domain. The digital information is then transferred back into the optical domain for layer $n+1$, for instance, via Mach-Zehnder modulator or optical attenuators. Such an approach is used by [15], [16].

In addition, one has to distinguish if the signal is detected with a direct or homo-/heterodyne detection scheme. In the latter one, the power of the local oscillator P_{LO} has to be taken with into account. The minimal input power that can be detected is noted as P_{\min} and depends on the detector's Noise-Equivalent Power (NEP) and its bandwidth.

The maximum numbers layers N for a given input power P_{in} of an OREO-based RNN, where each layer has an insertion loss α_{layer} , is summarized in

Table 2 Scalability of an OREO-based RNN. All powers in [dBm] and the insertion loss in [dB]. We assume the minimal signal power detectable with homo-/heterodyne detection to be $P_{\min}^{HD}(P_{LO}) \approx (P_{\min})^2/(2 P_{LO})$ (compare [17]).

	FF-arch	SL-arch. and ODO-arch.
Direct Detection	$N = \frac{(P_{\text{in}} - P_{\min})}{\alpha_{\text{layer}}}$	Independent of P_{in} as long as $P_{\text{in}} - \alpha_{\text{layer}} > P_{\min}$
Homo-/ Heterodyne detection	$N \propto \frac{(P_{\text{in}} - P_{\min}^{HD}(P_{LO}))}{\alpha_{\text{layer}}}$	Independent of P_{in} as long as $P_{\text{in}} - \alpha_{\text{layer}} > P_{\min}^{HD}(P_{LO})$

The current design of OREO has an intrinsic loss of 2 dB due to the used circulators plus 3 dB of the pulse depletion. Note that the latter value depends on the exact configuration of OREO (compare Figure 2 of the main text), and we assume that on average it represents the single pulse interaction (SPI) value. A complete layer of an RNN is formed if OREO is combined with a matrix multiplication and an nonlinear activation function, which we assume to have a loss of 1.32 dB as reported by[13]. Hence, total insertion loss per layer $IL_{\text{layer}} \approx 6.32$ dB. With the minimal detection power $P_{\min} \approx -24$ dBm of the used photodetector and the initial input power of $P_{\text{in}} \approx 0$ dBm, one could realize about four layers. Considering the current configuration of the OREO setup, one could realize up to four layers in an OREO-based RNN In comparison, to best of our knowledge, the maximum number of layers reported in an ONN without transferring information back to the digital domain is so far three[13], [14]

In order to increase the number of layers in the future of an OREO-based RNN, one could use a lower SIP-depletion value. For instance, one could convert only 10 % of the data pulse into the acoustic domain, yielding a loss of ≈ 0.5 dB. This would yield a maximum number of ≈ 6 layers that could be realized. Another two layers could be added by replacing the current Newport 12 GHz photodetectors with 10 GHz photodetectors manufactured by Thorlabs. Eventually, one can exchange direct detection with homo-/heterodyne detection which has been already used in [13]. In this case, one could push the maximum number of layers to approximate 16, considering a local oscillator power of $P_{LO} = 1$ mW.

Changes to the manuscript:

3. [Supplementary information] *Added section S4, discussing the scalability of OREO.*

4. [Main text, p. 15] Added the following text:

“Considering the current configuration of the OREO setup, one could realize up to four layers in an OREO-based RNN In comparison, to best of our knowledge, the maximum number of layers reported in an ONN without transferring information

back to the digital domain is so far three[13], [14]. A more detailed discussion on the scalability of an OREO-based RNN is provided by the supplementary material.”

Comment 9: “Optical power per control pulse of > 100 mW for photonic crystal fiber (PCF) is extremely high and will drive up power consumption even under the assumption that control pulses can be recycled. Are there any other waveguiding candidates that could operate with lower power control pulses? What are the estimates for control pulse power in case of integrated solutions?”

Response 9: The power consumption of OREO is mainly related to the control pulse power $P_{control}$ which is chosen to induce the SBS process., The underlying effective efficiency of a Brillouin process g_{eff} can be approximated as

$$g_{eff} = g L_{int} P_{control} - \alpha L$$

with the Brillouin gain g_{SBS} , the effective interaction length of the two pulses L_{int} , the optical loss α and the length of the Brillouin-active waveguide $L \geq L_{int}$.

By choosing different integrated platforms one effectively changes the values for g and α . For instance, the used PCF has a gain of $g^{PCF} \approx 2.5 \text{ m}^{-1}\text{W}^{-1}$ and a loss of $\alpha^{PCF} \approx 2.7 \text{ dB/km}$ [26]. In comparison, an on-chip chalcogenide waveguide can have a gain of $g^{AsS} \approx 500 \text{ m}^{-1}\text{W}^{-1}$ with a loss $\alpha^{AsS} \approx 0.5 \text{ dB/cm}$ [2]. In this case one can reduce the required optical power to generate the same g_{eff} by 99% to 1.27 mW. Recalling that this is a theoretical improvement and keeping in mind the experimental Brillouin demonstrations on-chip, we would frame this improvement as a long-term goal. Nonetheless, we would like to highlight the demonstration on a hybrid waveguides[27] with a gain of $g^{hybrid} = 750 \text{ m}^{-1}\text{W}^{-1}$ and the proposals[28] for sub-wavelength waveguides with gigantic Brillouin gain values $g > 10000 \text{ m}^{-1}\text{W}^{-1}$.

Changes to the manuscript:

We added a description on how on-chip platforms could improve the efficiency of OREO to the discussion section of the main text.

Comment 10: “ The authors claim that OREO might reach the accuracy of 97% (Line 253 and Supplementary Fig. 7). How does this compare with available prediction models?”

Response 10: We would like to thank the reviewer for asking for a comparison with available machine learning models. As suggested by the reviewer, we include in the revised manuscript a comparison of the OREO with a digital RNN. In the additional study, we found that RNN’s performance on the proposed pattern recognition task is almost on par with OREO.

Changes to the manuscript:

We added section S9 to the supplementary information to elaborate on the comparison of OREO with an RNN.

Comment 11: “In Supplementary material, Fig S3, the authors say that for B-E only acoustic wave b1 is observed and for F-I only b2. In the description of the procedure in section S4 it is said that the control pulse that is not swept is being kept constant. How does this eliminate the influence of the other acoustic wave? I would assume that aD1-aC1 interaction (yielding b1) will be there while aC2 is being swept. Could the authors clarify?”

Response 11: We would like to thank the reviewer for making the point. We need to clarify what we initially meant. First of all, we did not mean to give the impression that in Fig. S3 B-E only acoustic wave b_1 and in F-I only b_2 are presented. The reviewer is correct stating that while sweeping $a_{\{C,2\}}$ there is a contribution of b_1 . We admit that in the description of Fig. 3 F and I we did not include the influence of $a_{\{C, 2\}}$ on b_1 . We will clarify this in the following.

Changes to the manuscript:

1. [Supplementary material, p. 10]

Previously:

“As the acoustic control amplitude dictates the degree of depletion of the data pulse, we can use it to isolate the impact of certain acoustic contributions.”

Updated:

“The amplitude of $a_{\{C, i\}}$ influences the degree of depletion of the data pulse $a_{\{D, i\}}$, and therefore the level of the acoustic wave b_i which we create. Hence, we can change the amplitude of b_i by changing the amplitude of $a_{\{C, i\}}$. In addition, due to the coupling of the optical and acoustic field, previous generated acoustic waves b_k , $k < i$ also depend on $a_{\{C, i\}}$ after the Brillouin process yielding the dependence $b_k(a_{\{C, k\}}, a_{\{C, i\}})$.

Hence, we can use the interaction $a_{\{D, 1\}} \leftrightarrow a_{\{D, 1\}}$ to isolate the impact of b_1 on subsequent SBS processes if we sweep the amplitude $a_{\{C, 1\}}$.”

2. [Supplementary material, p. 11]

Previously:

“Next, the amplitude sweep of control pulse $a_{\{C, 2\}}$ (see Fig S3 F-I) reveals information about the influence of the acoustic wave b_2 .”

Updated:

“Next, the amplitude sweep of control pulse $a_{\{C, 2\}}$ (see Fig S3 F-I) reveals information about the influence of the acoustic wave b_1 and b_2 . Although we only change the amplitude of $a_{\{C, 2\}}$, the coupling of the optical and acoustic domain influences the level of b_1 as well.

3. [Supplementary material, Caption of Figure S3]

Previously:

“This measurement isolates the impact of the acoustic wave b_1 on subsequent SBS processes. [...]. This measurement isolates the impact of the acoustic wave b_2 .”

Updated:

“This measurement isolates the impact of the acoustic wave b_1 on subsequent SBS processes while b_2 and b_3 remain constant. [...]. This measurement isolates the combined impact of the acoustic wave b_1 and b_2 .”

Reply to Reviewer 3

Comment 1: *“In this work, the authors employ the theory of stimulated Brillouin scattering (SBS) to demonstrate optoacoustic recurrent operators (OREOs) which are very crucial for the short-term memory requirement of optical recurrent neural networks. In contrast to the rapid progress in optical deep neural networks utilizing either diffractive optics or photonic interferometric mesh in recent years, the optical analogy of recurrent neural network has not been well established. In this regard, the result of this work is very intriguing and promising for optical neuromorphic processing especially of temporal data.”*

Response 1: We would like to thank the reviewer for taking the time to evaluate the manuscript, for acknowledging the research gap that our work is addressing and for stressing its impact.

Comment 2: *“As being not familiar with SBS, the only concern for me is that the mechanism of SBS and its operation principle described in experimental results are quite complicated and not easy to catch up, while such in-depth understanding is essential to assess the importance of the manuscript.”*

Response 2: We would like to thank the reviewer for pointing out the need for additional clarification about the SBS dynamic. In order to do so, we addressed not only the subsequent comments of the reviewers but also added more changes to the manuscript. If required and specified, we are happy to add more.

Changes to the manuscript:

1. [Main text, p. 7]

Previously:

“In the following, we study the acoustic link by sweeping the data pulse amplitude of either $a_{\{D, 1\}}$ or $a_{\{D,2\}}$, while keeping the corresponding subsequent pulses constant. For instance, if the input amplitude of $a_{\{D, 1\}}$ is varied, $a_{\{D, 2\}}$ and $a_{\{D, 3\}}$ are fixed in amplitude. The control pulses $a_{\{C, i\}}$ are kept constant over the entire study. For each amplitude step, we measure the pulse’s area under curve (AuC) of the output pulses $a'_{\{D, i\}}$. An AuC-measurement of $a_{\{D, i\}}$ without control pulses serves as reference. In total, three different acoustic links occur from this experimental configuration, namely, $a_{\{D, 1\}} \rightarrow a'_{\{D, 2\}}$, $a_{\{D, 1\}} \rightarrow a'_{\{D, 3\}}$, and $a_{\{D, 2\}} \rightarrow a'_{\{D, 3\}}$ (see Figure A).”

Updated:

“In the following, we continue with the previously introduced setting of three pulse pairs ($a_{\{D, i\}}$, $a_{\{C, i\}}$), $i = 1, 2, 3$. We use this configuration to experimentally study the recurrent feedback, established by the acoustic waves b_i . Precisely, three different acoustic links occur. On the one hand, b_1 connects the input $a_{\{D, 1\}}$ to the output $a'_{\{D, 2\}}$ and $a'_{\{D, 3\}}$ and, on the other hand, b_2 connects $a_{\{D, 2\}}$ to $a'_{\{D, 3\}}$ (see Figure 2 A). In order to study the different connections, we sweep the input amplitude of either $a_{\{D, 1\}}$ or $a_{\{D, 2\}}$, while keeping the other data pulses constant. For instance, if the input amplitude of $a_{\{D, 1\}}$ is varied, $a_{\{D, 2\}}$ and $a_{\{D, 3\}}$ are fixed in amplitude. The control pulses $a_{\{C, i\}}$ are kept constant over the entire study. In the subsequent analysis, we normalize the AuC of the output pulses $a'_{\{D, i\}}$ with the AuC of an input data pulse $a_{\{D, i\}}$ with an amplitude scale of 1,

representing the highest input value. In order to measure the reference, we launch a data pulse sequence into OREO without counter-propagating control pulses.

2. [Main text, p. 8] Added a description of the single pulse interaction (SPI) to clarify its usage.
3. [Main text, p. 11] Updated the schematic illustration of OREO as acceptor in Figure 3A. Updated the figure caption to give more explanation.

Previously:

"As a measure of OREO's performance, we launch an evaluation pulse pair into the optical fiber and use the AuC of an output evaluation pulse (Eval) (see Figure 3 A). In total, we check all patterns 250-times in a random order and classify the resulting data set (70%training, 30%testing) with a Random Forest classifier (RFC) implemented in sklearn-package."

Updated:

"Figure 3 A shows schematically the information flow of the acceptor task. The individual pulses of the pattern ab are launched into OREO each after 2.5 ns with a corresponding control pulse $a_{C,i}$. The information of the pattern is captured by the acoustic wave and flows in time connecting the different SBS processes. Eventually, the sequential information captured by OREO is evaluated with a third optical evaluation pulse (Eval). In this way, we map the sequential information onto a single optical pulse which can be used then for classification, realizing experimentally a photonic acceptor. We perform the final classification by measuring the output evaluation pulse (Eval'), extracting its AuC in post-processing and then feeding this value into a digital RandomForest classifier (RFC). In total, we check all patterns 250-times in a random order and classify the resulting experimentally obtained data set (70 % training, 30 %testing) with the RFC implemented in sklearn-package."

Comment 3: *"When $a_{\{D,1\}}$ encounters $a_{\{C,1\}}$, they generate b_1 without any prior existence of acoustic waves. After then, does the subsequent SBS between $a_{\{D,2\}}$ and $a_{\{C,2\}}$ requires b_1 to generate b_2 ? It would be better to clarify whether any pair of $a_{\{D,i\}}$ and $a_{\{C,i\}}$ generates b_i regardless of prior b_j for $j < i$, and so what the difference is between SBS with and without prior b 's. While I could find the hints in the subsequent Fig. 2 (does this comparison correspond to line 151?) and supplementary information S6, in my opinion, it should be commented earlier.."*

Response 3: We thank the reviewer for pointing out the need for additional clarification about the requirement and influence of the different acoustic waves.

Firstly, the subsequent SBS process between $a_{\{D,2\}}$ and $a_{\{C,2\}}$ does not require the acoustic wave b_1 . However, the acoustic wave b_1 seeds the second Brillouin process and changes its properties. Secondly, in the submitted manuscript, we introduced the single pulse interaction (SPI), quantifying the degree of depletion when a single $a_{\{D,i\}}$ encounters $a_{\{C,i\}}$ without any acoustic seed. In this case, the degree of depletion depends mostly on the power of the control pulse. This changes as soon as an acoustic wave seeds a the SBS process. This seed can either enhance the strength of the Brillouin process or suppressing it depending on the phase of the acoustic wave. In other words, a first Brillouin process i generates an acoustic wave b_1 with a specific phase. The acoustic phase is the result of a detuned Brillouin process, meaning that the frequency difference between data and control matches

not exactly the Brillouin frequency Ω . The second Brillouin process $j > i$ generates a second acoustic wave b_2 on top of b_1 which interfere. In the case of a matched Brillouin process this interference would always be constructive and amplify the Brillouin process. As consequence, the nonlinear dynamic in Figure 2 B-D. Note, that we have discussed the matched case in the supplementary material (e.g., see Figure S.6). As Brillouin scattering describes the dynamic coupling between acoustic and optical domain, the acoustic wave has an immediate effect on the coupling. For instance, if the b_1 and b_2 interfere destructively, then reaching the “stimulated” SBS is slowed down and therefore the depletion of the data pulse $a_{D,2}$ weaker. However, if b_1 and b_2 add up via constructive interference, the “stimulated” regime is reached earlier and more of data pulse $a_{D,2}$ is depleted. We added details to this to the revision.

Changes to the manuscript:

1. [Main text, p. 7]

Previously:

“Because the degree of depletion is lower as for a single pulse interaction (SPI), e.g., $a_{D,1} \rightarrow a'_{D,3}$, we conclude that the acoustic wave b_1 weakens the SBS process of $a_{D,2} \leftrightarrow a_{C,2}$.”

Updated:

“In order to have a reference for the dynamic of the Brillouin process, we use the interaction of a single data-control pulse pair. The data pulse a_D gets depleted by the SBS process in this single pulse interaction (SPI). The degree of depletion depends mainly on the power carried by the control pulse[19]. For OREO, we deplete about 47 % of the data pulse.”

2. [Main text, p. 8] Added the following text block:

“For example, if the interaction $a_{D,i} \leftrightarrow a_{C,i}$ has created the acoustic wave b_i with a specific phase (see Equation (2)), then a second SBS process $j > i$ would generate dynamically b_j on top of b_i . Both b_i and b_j have a different phase which lead to acoustic interference. As Brillouin scattering describes a dynamic coupling between photons and phonons, the acoustic interference effects the overall “stimulated” dynamic. Hence, if b_i and b_j interfere destructively, “stimulated” dynamic is slowed down and, therefore, the depletion of the data pulse $a_{D,j}$ lower. However, if b_i and b_j add up via constructive interference, then stimulation is accelerated, increasing the depletion of $a_{D,j}$.”

Comment 4: “Typo in Fig. 1A: the final weight matrix seems to be W^{N-1} .”

Response 4: We would like to thank the reviewer for pointing out the spelling mistake.

Changes to the manuscript:

[Main text, Fig. 1A] Changed W^N to W^{N-1} .

Comment 5: “Two sentences are very confusing: “b1 weakens...” in line 152 and “b1 enhances...” in line 157.”

Response 5: We would like to thank the reviewer for pointing out additional need for clarification. We revised the main text between the suggested lines 152 to 157, in order to clarify the underlying dynamic.

Changes to the manuscript:

[Main text, p. 8]

Previously:

“Because the degree of depletion is lower as for SPI-case, e.g., $a_{D,1} \rightarrow a'_{D,3}$, we conclude that the acoustic wave b_1 weakens the SBS process of $a_{D,2} \leftrightarrow a_{C,2}$.”

Updated:

“The reason for this dynamic is that $a_{D,2} \leftrightarrow a_{C,2}$ is seeded by the acoustic wave b_1 influencing the degree of depletion. More precisely, the dynamic can be explained with the different acoustic phases of b_1 and b_2 (see equation (2)) which can lead to constructive or destructive interference during the SBS process. The acoustic phase is introduced by detuning the frequency difference between data and control pulses slightly from the Brillouin frequency.

For example, if the interaction $a_{D,i} \leftrightarrow a_{C,i}$ has created the acoustic wave b_i with a specific phase, then a second SBS process $j>i$ would generate dynamically b_j on top of b_i . Both b_i and b_j have a specific phase which lead to acoustic interference.

As Brillouin scattering describes a dynamic coupling between photons and phonons, the acoustic interference effects the overall “stimulated” dynamic. Hence, if b_i and b_j interfere destructively, “stimulated” dynamic is slowed down and, therefore, the depletion of the data pulse $a_{D,j}$ lower. However, if b_i and b_j add up via constructive interference, then stimulation is accelerated, increasing the depletion of $a_{D,j}$.

The acoustic interference is also the reason for the decreasing behavior of the link $a_{D,1} \rightarrow a'_{D,3}$ on $a_{D,3} \leftrightarrow a_{C,3}$. Here, the acoustic wave b_1 accelerates the SBS process of $a_{D,3} \leftrightarrow a_{C,3}$ which leads to an increased depletion of $a_{D,3}$.”

Comment 6: “Is there any possibility of interactions for data/control pulses $a_{D,i}$ and $a_{C,j}$ for different i and j at different position?”

Response 6: We like to thank the reviewer for raising the question about multiple in space separated OREOs. Indeed, as briefly mentioned in supplementary section S1, one is not limited to the presented interaction scheme. The proposed interaction could be realized if one uses a longer waveguide. Assuming that the data pulses $a_{D,i}$ are traveling from left to right and vice versa for the control pulses. Then, one could perform a temporal calibration in a way that $a_{D,1}$ could interact with $a_{C,1}$ at the left edge of the waveguide and with $a_{C,2}$ at the right side.

Comment 7: “The details of the RFC and the way how it is trained could be described in Methods section.”

Response 7: We would like to thank the reviewer for pointing out the missing information about the RFC. We will add the information.

Changes to the manuscript:

[Main text, p. 20] Added subsection to Method section.

“We use a RandomForestClassifier (RFC) to classify the area under curve of the output evaluation pulses Eval'. We initialize our RFC with the sklearn python package, setting the maximal depth to 10 and the random state to 2. We found those to be the configuration of the RFC to achieve the highest accuracies for both the ab and the abc-study. The function `textscsklearn.model_selection.train_test_split` splits the initial data sets into the test and training one with a random state of 20. We ensure that that the different patterns are equally distributed to the train and test data set with the stratify keyword of the `textstrain_test_split`-function. We fit the RFC to the data set using its default fit-function.”

References

- [1] S. Buddhiraju, A. Dutt, M. Minkov, I. A. D. Williamson, and S. Fan, “Arbitrary linear transformations for photons in the frequency synthetic dimension,” *Nat Commun*, vol. 12, no. 1, p. 2401, Dec. 2021, doi: 10.1038/s41467-021-22670-7.
- [2] B. J. Eggleton, C. G. Poulton, P. T. Rakich, Michael. J. Steel, and G. Bahl, “Brillouin integrated photonics,” *Nat. Photonics*, vol. 13, no. 10, pp. 664–677, Oct. 2019, doi: 10.1038/s41566-019-0498-z.
- [3] M. Merklein, I. V. Kabakova, A. Zarifi, and B. J. Eggleton, “100 years of Brillouin scattering: Historical and future perspectives,” *Applied Physics Reviews*, vol. 9, no. 4, p. 041306, Dec. 2022, doi: 10.1063/5.0095488.
- [4] Y. Liu *et al.*, “Circulator-Free Brillouin Photonic Planar Circuit,” *Laser & Photonics Reviews*, vol. 15, no. 5, p. 2000481, May 2021, doi: 10.1002/lpor.202000481.
- [5] F. A. Gers, “Learning to forget: continual prediction with LSTM,” in *9th International Conference on Artificial Neural Networks: ICANN '99*, Edinburgh, UK: IEE, 1999, pp. 850–855. doi: 10.1049/cp:19991218.
- [6] R. C. Staudemeyer and E. R. Morris, “Understanding LSTM -- a tutorial into Long Short-Term Memory Recurrent Neural Networks.” arXiv, Sep. 12, 2019. Accessed: Jan. 15, 2024. [Online]. Available: <http://arxiv.org/abs/1909.09586>
- [7] S. Becker, A. Geilen, and B. Stiller, “High-speed coherent photonic random-access memory in long-lasting sound waves.” arXiv, Nov. 10, 2023. Accessed: Nov. 13, 2023. [Online]. Available: <http://arxiv.org/abs/2311.06219>
- [8] Q. V. Le, N. Jaitly, and G. E. Hinton, “A Simple Way to Initialize Recurrent Networks of Rectified Linear Units.” arXiv, Apr. 07, 2015. Accessed: Jan. 08, 2024. [Online]. Available: <http://arxiv.org/abs/1504.00941>
- [9] N. Zucchet, R. Meier, S. Schug, A. Mujika, and J. Sacramento, “Online learning of long-range dependencies.” arXiv, Nov. 06, 2023. Accessed: Jan. 08, 2024. [Online]. Available: <http://arxiv.org/abs/2305.15947>
- [10] D. Y. Fu, T. Dao, K. K. Saab, A. W. Thomas, A. Rudra, and C. Ré, “Hungry Hungry Hippos: Towards Language Modeling with State Space Models.” arXiv, Apr. 28, 2023. Accessed: Jan. 08, 2024. [Online]. Available: <http://arxiv.org/abs/2212.14052>

- [11] Y. Goldberg, *Neural network methods in natural language processing*. in Synthesis lectures on human language technologies, no. 37. Cham: Springer nature Switzerland AG, 2022.
- [12] J. L. Elman, "Finding Structure in Time," *Cognitive Science*, vol. 14, no. 2, pp. 179–211, Mar. 1990, doi: 10.1207/s15516709cog1402_1.
- [13] S. Bandyopadhyay *et al.*, "Single chip photonic deep neural network with accelerated training." arXiv, Aug. 02, 2022. Accessed: Dec. 01, 2022. [Online]. Available: <http://arxiv.org/abs/2208.01623>
- [14] F. Ashtiani, A. J. Geers, and F. Aflatouni, "An on-chip photonic deep neural network for image classification," *Nature*, vol. 606, no. 7914, pp. 501–506, Jun. 2022, doi: 10.1038/s41586-022-04714-0.
- [15] Y. Shen *et al.*, "Deep learning with coherent nanophotonic circuits," *Nature Photon*, vol. 11, no. 7, pp. 441–446, Jul. 2017, doi: 10.1038/nphoton.2017.93.
- [16] H. H. Zhu *et al.*, "Space-efficient optical computing with an integrated chip diffractive neural network," *Nat Commun*, vol. 13, no. 1, p. 1044, Feb. 2022, doi: 10.1038/s41467-022-28702-0.
- [17] T. Iwasaki and T. Nemoto, "A Homodyne Detection at 100 GHz with a Pyroelectric Detector," *IEEE Trans. Instrum. Meas.*, vol. 29, no. 3, pp. 190–192, 1980, doi: 10.1109/TIM.1980.4314904.
- [18] Y. R. Bawankar and A. Singh, "Microring Resonators Based Applications in Silicon Photonics - A Review," in *2021 5th Conference on Information and Communication Technology (CICT)*, Kurnool, India: IEEE, Dec. 2021, pp. 1–6. doi: 10.1109/CICT53865.2020.9672427.
- [19] Z. Zhu, D. J. Gauthier, and R. W. Boyd, "Stored Light in an Optical Fiber via Stimulated Brillouin Scattering," *Science*, vol. 318, no. 5857, pp. 1748–1750, Dec. 2007, doi: 10.1126/science.1149066.
- [20] B. Stiller *et al.*, "Cross talk-free coherent multi-wavelength Brillouin interaction," *APL Photonics*, vol. 4, no. 4, p. 040802, Apr. 2019, doi: 10.1063/1.5087180.
- [21] G. Agrawal, "Stimulated Brillouin Scattering," in *Nonlinear Fiber Optics*, Elsevier, 2013, pp. 353–396. doi: 10.1016/B978-0-12-397023-7.00009-7.
- [22] Y. Goldberg, *Neural Network Methods for Natural Language Processing*. in Synthesis Lectures on Human Language Technologies. Cham: Springer International Publishing, 2017. doi: 10.1007/978-3-031-02165-7.
- [23] G. Weiss, Y. Goldberg, and E. Yahav, "On the Practical Computational Power of Finite Precision RNNs for Language Recognition." arXiv, May 13, 2018. Accessed: Nov. 27, 2023. [Online]. Available: <http://arxiv.org/abs/1805.04908>
- [24] B. Stiller *et al.*, "Brillouin light storage for 100 pulse widths." arXiv, Aug. 02, 2023. Accessed: Sep. 20, 2023. [Online]. Available: <http://arxiv.org/abs/2308.01009>
- [25] C. C. Rodrigues, R. O. Zurita, T. P. M. Alegre, and G. S. Wiederhecker, "Stimulated Brillouin scattering by surface acoustic waves in lithium niobate waveguides," *J. Opt. Soc. Am. B*, vol. 40, no. 5, p. D56, May 2023, doi: 10.1364/JOSAB.482656.
- [26] J.-C. Beugnot *et al.*, "Complete experimental characterization of stimulated Brillouin scattering in photonic crystal fiber," *Opt. Express*, vol. 15, no. 23, p. 15517, Nov. 2007, doi: 10.1364/OE.15.015517.
- [27] B. Morrison *et al.*, "Compact Brillouin devices through hybrid integration on silicon," *Optica*, vol. 4, no. 8, p. 847, Aug. 2017, doi: 10.1364/OPTICA.4.000847.
- [28] P. T. Rakich, C. Reinke, R. Camacho, P. Davids, and Z. Wang, "Giant Enhancement of Stimulated Brillouin Scattering in the Subwavelength Limit," *Phys. Rev. X*, vol. 2, no. 1, p. 011008, Jan. 2012, doi: 10.1103/PhysRevX.2.011008.

Reviewer #1 (Remarks to the Author):

I'm afraid that the fundamental concerns I had for publishing this paper in Nature Communications are still very much unaddressed. The changes to the manuscript in response to my comments are rather minor, and mostly come down to postponing the research for future work. While this is perfectly fine for other journals, I believe this does not meet the bar for Nature Communications, given that the current impact is low and the future impact is still very much in doubt...

Reviewer #2 (Remarks to the Author):

I would like to thank the authors for addressing all my comments and questions in a very detailed and convincing manner. The novelty and practicality of the proposed OREO is much better highlighted.

Adding benchmarking against digital RNNs and the study on path to integration have significantly strengthened the authors' case.

In my opinion, the manuscript can be published in its current form.

Reviewer #3 (Remarks to the Author):

The manuscript has been improved in terms of clarity. I have no further questions and now recommend it for publication.

One minor point: (Line 63) Blue arrows seem to appear in Fig. 1B, not in Fig. 1A. (black arrows in Fig. 1A)

Reply to Reviewer 3

Comment 1: Blue arrows seem to appear in Fig. 1B, not in Fig. 1A. (black arrows in Fig. 1A)

Response 1: We thank the reviewer for taking the time to evaluate our manuscript a second time. We put the note on the blue arrow to the description of Figure 1B.